# Mind the Gap: a Spectral Analysis of Rank Collapse and Signal Propagation in Attention Layers

**Thiziri Nait Saada** [* 1]   **Alireza Naderi** [* 1]   **Jared Tanner** [1]

## Abstract

Attention layers are the core component of transformers, the current state-of-the-art neural network architecture. Alternatives to softmax-based attention are being explored due to its tendency to hinder effective information flow. Even *at initialisation*, it remains poorly understood why the propagation of signals and gradients through these random networks can be pathological, resulting in issues known as (i) vanishing/exploding gradients and (ii) rank collapse *in depth*, i.e. when all tokens converge to a single representation along layers. While rank collapse in depth naturally arises from repeated matrix multiplications—a common pattern across various architectures—we identify an additional and previously unknown challenge unique to softmax attention layers: (iii) rank collapse *in width*, which occurs as the context length increases. Using Random Matrix Theory, we conduct a rigorous analysis that uncovers a spectral gap between the two largest singular values of the attention matrix as the cause of (iii), which in turn exacerbates (i) and (ii). Building on this insight, we propose a novel yet simple practical solution to mitigate rank collapse in width by removing the outlier eigenvalue(s). Our theoretical framework offers a fresh perspective on recent practical studies, such as (Ye et al., 2025; Ali et al., 2023), whose ad hoc solutions can now be interpreted as implicit efforts to address the spectral gap issue. This work provides valuable theoretical support for ongoing large-scale empirical research, bringing theory and practice one step closer in the understanding of transformers.[1]

[*]Equal contribution  [1]Mathematical Institute, University of Oxford. Correspondence to: Thiziri Nait Saada and Alireza Naderi <naitsaadat@maths.ox.ac.uk, naderi@maths.ox.ac.uk>.

*Proceedings of the 42nd International Conference on Machine Learning*, Vancouver, Canada. PMLR 267, 2025. Copyright 2025 by the author(s).

[1]Our code is available at https://github.com/thizirinait/Spectral-Analysis-of-Attention-Layers.git.

## 1. Introduction

Transformers (Vaswani et al., 2017) have emerged as the dominant architecture in machine learning, achieving remarkable success across various domains, particularly in natural language processing and computer vision, largely due to their defining feature: the self-attention mechanism (Bahdanau et al., 2016). However, despite their empirical success, transformers are often plagued by training instability and high sensitivity to numerous hyperparameters, which require careful tuning. This challenge has motivated recent efforts to establish a theoretical framework for understanding transformer architectures, even in their most basic forms, to ensure reliable information flow through deeper layers and facilitate training.

The purpose of this work is to analyse signal propagation in softmax-based attention layers *at initialisation*, i.e. with randomly initialised model parameters. While the issues of rank collapse (in depth) and vanishing/exploding gradients have been previously identified in transformers at initialisation (Dong et al., 2021; Noci et al., 2022), our work extends these findings and uncovers an additional phenomenon—rank collapse *in width*—due to the use of softmax in the self-attention mechanism. By width, we specifically refer to the context length, which, despite its increasing scale in modern attention layers, seems to have been overlooked in favour of depth in previous analyses of rank collapse in transformers. Indeed, rank collapse in width, which is unique to attention layers, has not been identified in the existing literature nor recognised as a catalyst for rank collapse along the depth. In contrast, rank collapse in depth is not specific to transformer-like networks, as it typically results from successive matrix multiplications, a shared feature of various deep neural network architectures. Investigating the spectra of the random matrices formed by the model's parameters, we detect a *spectral gap* between the two largest eigenvalues of the attention matrix, which drives rank collapse in width and further accelerates rank collapse in depth. Moreover, we propose a provably effective remedy for the spectral gap, a solution that naturally arises when the problem is viewed through a spectral lens. To the best of our knowledge, this is the first study to examine the attention mechanism in the large context length regime, as well as the first to identify

rank collapse in width and its effect on other oversmoothing phenomena.

Let us consider the eigenvalues of a softmax-based attention matrix. Since the rows sum to unity, there is an eigenvalue of 1 corresponding to the eigenvector of all-ones. Under certain conditions, the other eigenvalues shrink in size as the matrix dimension increases, resulting in a widening gap between the largest eigenvalue (i.e. 1) and the diminishing bulk of eigenvalues; see Figure 1. The successive multiplication of such matrices at each layer increasingly favours a specific direction—the one aligned with the dominant eigenvector of the attention matrix—over the others. This leads to a distortion in the geometry of the input training data, exemplified by the phenomenon of rank collapse. An intuitive solution is to project out this troublesome direction from all attention matrices, allowing for more balanced signal propagation. These ideas lie at the core of our rigorous analysis of attention layers, from which we derive a slightly modified attention mechanism that proves advantageous in practice. Remarkably, the spectral gap observed in softmax attention matrices also appears in alternative attention mechanisms suggested in the literature, making our analysis relevant to these variants as well; see Figure 1.

## 1.1. Related work

Our analysis builds upon previous work that studied the spectra of large random neural networks to better understand and stabilise initial training dynamics, such as (Pennington et al., 2018) for fully-connected networks and (Xiao et al., 2018) for convolutional networks.

The rank collapse in transformers was first explored in (Dong et al., 2021), where the authors show that the output of an attention-only transformer converges exponentially with depth to a single representation across tokens. (Noci et al., 2022) make a connection between rank collapse and vanishing gradients by assuming "uniform attention" $\mathbf{A} = \frac{1}{T}\mathbf{1}_{T \times T}$, essentially proving rank collapse in depth based on the assumption that rank collapse in width has already occurred (as opposed to showing why this premise holds).

There is also a growing literature on the shortcomings of softmax self-attention mechanism and possible alternatives. Some suggest the usage of other well-known activations such as identity, ReLU, and sigmoid (Hron et al., 2020; Wortsman et al., 2023; Ramapuram et al., 2025), while others propose modifications to the attention block (He et al., 2022; Wang et al., 2022; Shi et al., 2022; Ali et al., 2023; Dovonon et al., 2024; Ye et al., 2025). Based on our analysis, we can offer a fresh perspective on these attention variants and reflect on their effect on rank collapse. Indeed, they all fall (at the exception of the identity) within the class of attention matrices presenting a spectral gap, therefore

suffering from rank collapse, with slightly adjusted rates according to the location of the outlier in the spectrum; see Figure 1.

Interestingly, our work and the fix we propose find echos in recent promising practical studies. For instance, the Differential Transformer (Ye et al., 2025) computes the attention matrix as a difference between two softmax matrices and has shown significant improvements from this adjustment, e.g. finer quantisation, better key information retrieval or hallucination mitigation. In light of our findings, this adjustment can be understood as removing the troublesome outlier from the spectrum of the attention matrix that naturally arises when using softmax. Even closer, the work of (Ali et al., 2023) directly implements our fix and shows improved behaviour on image segmentation tasks. When our work is of theoretical nature and concerned with the initialisation dynamics, it is interesting to see how these concepts are playing out in practical applications.

## 1.2. Organisation of the paper

In Section 2, we first review the fundamentals of random matrix spectra before exploring the properties of standard softmax key-query attention matrices with isotropic input. We prove the existence of a spectral gap for such attention matrices and use this result to demonstrate rank collapse in width. Next, we introduce a modified attention mechanism designed to eliminate the spectral gap and show that this modification effectively resolves rank collapse in width.

In Section 3, we extend our analysis from a single attention layer to a deep attention-only transformer. Using a certain class of random Markov matrices to model attention, we precisely determine the rate at which rank collapse occurs as a function of width and depth (Proposition 3.4), thereby revealing for the first time how the two interact. Furthermore, we prove that the removal of the spectral gap cures rank collapse in width and mitigates the exploding gradients (Propositions 3.7 and 3.8).

In Section 4, we validate our findings, providing empirical evidence of rank collapse in both width and depth in commercial models such as BERT family. We put to test our "remove the gap" solution across a range of architectures featuring LayerNorm and skip connections and show its effectiveness in resolving stable rank collapse and slowing down gradient explosion. Finally, we observe the emergence of additional outliers in deep transformers and conduct experiments to study the potential benefits of removing all outliers rather than the dominant one.

## 2. Attention layers and rank collapse in width

### 2.1. Spectra of random matrices

Throughout this paper, we consider random matrices (of different distributions) in the *large width* limit and describe them through their *limiting* spectral properties. In the context of transformers, we mean by "large width" that both the number of tokens and the embedding dimension(s) are large—an assumption typically satisfied in practice. For certain classes of random matrices, the overall behaviour of eigenvalues/singular values becomes remarkably predictable as the matrix size increases, despite the randomness of individual entries. If $\mathbf{M}_n \in \mathbb{R}^{n \times n}$ are random matrices with eigenvalues and singular values denoted by $\{\lambda_i(\mathbf{M}_n)\}$ and $\{s_i(\mathbf{M}_n)\}$, respectively, the histograms of the $n$ eigenvalues/singular values

$$\mu_{\mathbf{M}_n} := \frac{1}{n} \sum_{i=1}^n \delta_{\lambda_i(\mathbf{M}_n)}, \quad \nu_{\mathbf{M}_n} := \frac{1}{n} \sum_{i=1}^n \delta_{s_i(\mathbf{M}_n)},$$

converge, in many interesting cases, to deterministic distributions $\mu$ and $\nu$, known as the *limiting eigenvalue/singular value distribution* of $\mathbf{M}_n$. Additionally, the *largest* eigenvalue/singular value of random matrices is often studied in its own right. Our analysis builds on cutting-edge results concerning both the limiting distribution of the eigenvalues/singular values (the "bulk" of the spectrum) and the behaviour of the largest eigenvalue/singular value (the "edge" of the spectrum). In particular, we focus on two classes of random matrices: Gaussian and Markov, that respectively model the *value* and *attention* matrices in our transformer model (11) at initialisation.

### 2.2. Attention at initialisation

Let $\mathbf{X}_0 \in \mathbb{R}^{T \times d}$ be an input sequence of context length $T$, i.e. formed of $T$ tokens. Each token is embedded as a $d$-dimensional vector, with a fixed ratio $\gamma := \frac{T}{d} \leq 1$. The standard softmax key-query attention matrix is obtained by

$$\mathbf{A}(\mathbf{X}_0) := \mathrm{softmax}\left(\frac{\mathbf{X}_0 \mathbf{W}^Q \mathbf{W}^{K\top} \mathbf{X}_0^\top}{\sqrt{d_{qk}}}\right), \quad (1)$$

where $\mathbf{W}^Q, \mathbf{W}^K \in \mathbb{R}^{d \times d_{qk}}$ have i.i.d. $\mathcal{N}(0, \sigma_{qk}^2)$ entries.

The above attention matrix exhibits a spectral gap, meaning its largest eigenvalue (or singular value) differs significantly in magnitude from the second largest. Moreover, this gap widens as the context length increases. In fact, we precisely quantify the rates at which both eigenvalues evolve, as follows.

**Theorem 2.1** (Spectral gap in softmax attention). *Let the input sequence $\mathbf{X}_0$ have orthonormal rows, i.e. $\mathbf{X}_0 \mathbf{X}_0^\top = \mathbf{I}$. Then, $\lambda_1(\mathbf{A}(\mathbf{X}_0)) = 1$ and almost surely,*

$$\lim_{T \to \infty} s_1(\mathbf{A}(\mathbf{X}_0)) = 1, \quad (2)$$

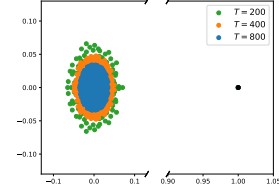
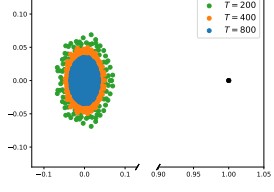

(a) Softmax attention (Vaswani et al., 2017)  (b) i.i.d. Markov

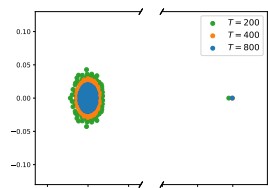
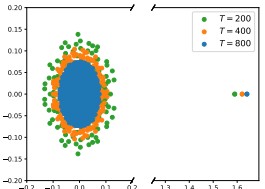

(c) ReLU attention (Wortsman et al., 2023)  (d) Sigmoid attention (Ramapuram et al., 2025)

*Figure 1.* Theorem 2.1 demonstrates that applying the conventional softmax key-query attention mechanism to orthonormal input tokens yields a spectral gap in its spectrum (a). Due to their non-zero mean, ReLU (c) and sigmoid attention (d) mechanisms also exhibit a spectral gap—although of a different size. As the size $T$ of an i.i.d. Markov matrix (Definition 3.1) grows, its eigenvalues form a circular bulk of radius $O(T^{-1/2})$ in the complex plane, except for the largest eigenvalue which remains 1 (the black dot in (b)). In practice, $T$ does not need to be too large for the limiting behaviour to appear, as shown above.

*and*

$$\lim_{T \to \infty} s_2(\sqrt{T} \mathbf{A}(\mathbf{X}_0)) = 2\sigma, \quad (3)$$

*while*

$$\overline{\lim}_{T \to \infty} |\lambda_2(\sqrt{T} \mathbf{A}(\mathbf{X}_0))| \leq 2\sigma, \quad (4)$$

*where $\sigma$ depends only on $\sigma_{qk}$.*

With no further assumptions, we exactly analyse one attention layer at initialisation, where the output is given by

$$\mathbf{X} = \mathbf{A}(\mathbf{X}_0) \mathbf{W}^V,$$

and the *value matrix* $\mathbf{W}^V \in \mathbb{R}^{d \times d}$ is initialised independently with i.i.d. $\mathcal{N}(0, 1)$ entries [2].

### 2.3. Rank collapse *in width*

Theorem 2.1 reveals that the attention matrix becomes effectively rank-one for large context length $T$. Naturally, any definition of "rank collapse" relies on a proxy for discrete rank, as the random matrices in question are almost surely full-rank, making it uninformative to refer to their

---

[2]See Remark 3.3 for a discussion on scaling the value matrix.

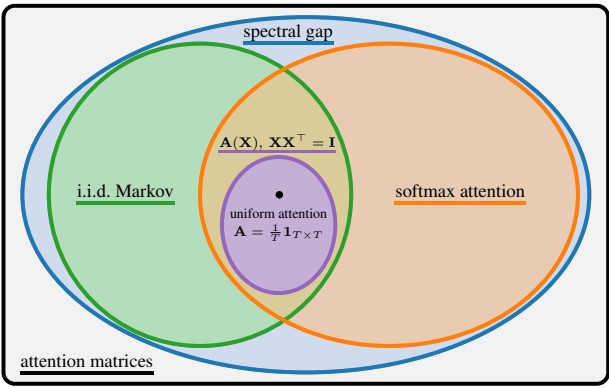

*Figure 2.* The standard key-query attention matrices of Equation (1) form a subclass of i.i.d. Markov matrices of Definition 3.1, of which the uniform attention of (Noci et al., 2022) is one trivial special case. We use i.i.d. Markov matrices to build our general multi-layer theory in section 3.

actual rank. For example, (Dong et al., 2021) consider the "one-infinity norm" of the residual (the difference between a matrix and its best approximation of the form $\mathbf{1}\mathbf{x}^\top$), defined as $\sqrt{\|\text{res}(\mathbf{M})\|_1 \|\text{res}(\mathbf{M})\|_\infty}$, while (Noci et al., 2022) use $\sum_{i,j}(\mathbf{M}\mathbf{M}^\top)_{i,j}$, which is maximised when all rows of $\mathbf{M}$ are identical. We choose *stable rank* as our preferred proxy due to its clear geometrical interpretation and simple definition in terms of singular values. It is defined for any non-zero $\mathbf{M} \in \mathbb{R}^{m \times n}$ as

$$\text{sr}(\mathbf{M}) := \frac{\|\mathbf{M}\|_F^2}{\|\mathbf{M}\|^2} = \frac{\sum_i s_i^2(\mathbf{M})}{s_1^2(\mathbf{M})}. \qquad (5)$$

The following theorem establishes that the stable rank of the covariance kernel

$$\boldsymbol{\Sigma} := \mathbf{X}\mathbf{X}^\top$$

collapses to 1 as the width $T$ increases.

**Theorem 2.2** (Rank collapse in width). *Assume* $\boldsymbol{\Sigma}_0 = \mathbf{I}$. *Then, with overwhelming probability*[3],

$$\lim_{T \to \infty} \text{sr}(\boldsymbol{\Sigma}) = 1. \qquad (6)$$

*Moreover, the convergence happens at a polynomial rate, i.e.* $\left|\text{sr}(\boldsymbol{\Sigma}) - 1\right| = O(T^{-3})$.

While the isometric input assumption may appear restrictive, it is introduced solely to formalise the theoretical argument. In Appendix A.1, we present extensive empirical evidence that this assumption is not required for the result to hold in practice.

---

[3]A sequence of events $E_n$ holds with overwhelming probability if, for every $A > 0$, $\mathbb{P}(E_n) \geq 1 - C_A n^{-A}$, for some constant $C_A$, as defined in (Tao, 2012).

This theorem constitutes the primary contribution of this paper. It uncovers a previously unknown phenomenon that we refer to as rank collapse in width—or the convergence of tokens' representations into a single point as the context length increases within a single attention layer. It should not be mistaken for rank collapse in depth, which is partly a result of repeated random matrix multiplications. In contrast, rank collapse in width is unique to attention layers, making this discovery particularly significant for the theory of transformers. In the next section, we will explore how removing the spectral gap of Theorem 2.1 helps alleviate rank collapse in width and restore a more balanced flow of information.

### 2.4. Attention without the gap

Assume $\mathbf{X}_0\mathbf{X}_0^\top = \mathbf{I}$ and let us write $\mathbf{A} := \mathbf{A}(\mathbf{X}_0)$ to lighten the notation. The attention matrix $\mathbf{A} \in \mathbb{R}^{T \times T}$ can be written as

$$\mathbf{A} = \mathbb{E}\mathbf{A} + (\mathbf{A} - \mathbb{E}\mathbf{A}) = \frac{1}{T}\mathbf{1}_{T \times T} + \mathbf{A}^\perp, \qquad (7)$$

where $\mathbf{1}_{T \times T} := (1, \cdots, 1)^\top(1, \cdots, 1)$ is the all-ones matrix. Therefore, $\mathbf{A}$ is a rank-one perturbation of $\mathbf{A}^\perp$, whose spectral radius is $\lambda_1(\mathbf{A}^\perp) = O(T^{-1/2})$ (see Lemma A.3). Although the rank-one perturbation $\frac{1}{T}\mathbf{1}_{T \times T}$ cannot disturb the bulk of the spectrum, it causes the largest eigenvalue to "escape" from the bulk to 1, creating a spectral gap.

In light of this, we can slightly modify the attention mechanism to eliminate the outlier—and thus the gap—simply by replacing $\mathbf{A}$ with $\mathbf{A}^\perp$ to get the following modified attention layer, i.e.

$$\mathbf{X}^\perp = \mathbf{A}^\perp(\mathbf{X}_0)\mathbf{W}^V, \qquad (8)$$

where $\mathbf{A}^\perp(\mathbf{X}_0) := \mathbf{A}(\mathbf{X}_0) - \frac{1}{T}\mathbf{1}_{T \times T}$. Note that this modification is applied only to the attention matrix (and *not* to the signal representation) and $\mathbf{X}^\perp$ serves as shorthand for the output of an attention layer whose $\mathbf{A}$'s is replaced with $\mathbf{A}^\perp$'s as in (8).

Since the modified attention exhibits no spectral gap (see Lemma A.6), the stable rank of the covariance matrix $\boldsymbol{\Sigma}^\perp := \mathbf{X}^\perp{\mathbf{X}^\perp}^\top$ no longer collapses to 1 but instead scales linearly in width, as detailed in Proposition 2.3 (cf. Theorem 2.2).

**Proposition 2.3** (Resolved rank collapse in width). *Let* $\mathbf{X}^\perp = \mathbf{A}^\perp(\mathbf{X}_0)\mathbf{W}^V$ *be the signal in our modified attention layer* (8) *and* $\boldsymbol{\Sigma}^\perp \in \mathbb{R}^{T \times T}$ *be its covariance matrix. Then, almost surely, the rank does not collapse, i.e., there exists a constant* $C > 0$ *such that,*

$$\lim_{T \to \infty} \frac{\text{sr}(\boldsymbol{\Sigma}^\perp)}{T} = C. \qquad (9)$$

## 3. The interplay between width and depth

It is important to note that a spectral gap within the attention matrix can arise when the input to the softmax function consists of a matrix with vanishing elements. This can be achieved, for example, by initialising the key and query matrices following Xavier or He initialisation schemes, or under any scaling such that $\sigma_{qk}^2 \to 0$ as $d_{qk}$ increases. In this scenario, the attention matrix asymptotically approaches the uniform attention matrix, i.e. the matrix of all entries equal to $1/T$. In this edge case, the attention matrix becomes rank-one, the spectral gap is maximised and once the rank-one perturbation $\frac{1}{T}\mathbf{1}_{T \times T}$ is removed, the attention is reduced to zero—effectively preventing any signal propagation beyond the rank collapse.

It is, nonetheless, in this trivial case that the analysis of signal propagation along depth was conducted in (Noci et al., 2022). In particular, their study demonstrates rank collapse in depth by implicitly assuming that rank collapse in width had already occurred. In contrast, our work presents a more refined analysis of rank collapse by considering the effects of both width and depth jointly, offering, to the best of our knowledge, the most advanced theoretical study to date on the initialisation of transformers.

We model the $\ell$-th layer attention mechanism at initialisation by a random matrix $\mathbf{A}_\ell$ with non-negative entries $(\mathbf{A}_\ell)_{i,j} \geq 0$ and normalised rows, i.e. $\sum_j (\mathbf{A}_\ell)_{i,j} = 1$, as if it were generated by a row-wise application of softmax. As we will demonstrate, this model functions as a helpful abstraction that offers insights into the interaction of width and depth. More specifically, we consider $\mathbf{A}_\ell$ to be an i.i.d. Markov matrix, as defined in (Bordenave et al., 2011).

**Definition 3.1** (i.i.d. Markov matrix). Let $Z_{i,j}$ be i.i.d. non-negative random variables with positive mean $m := \mathbb{E}(Z_{1,1}) > 0$ and variance $\sigma^2 := \mathrm{Var}(Z_{1,1}) > 0$ as well as finite fourth moment $\mathbb{E}(Z_{1,1}^4) < \infty$. Let $\mathbf{A} \in \mathbb{R}^{T \times T}$ be its row-normalised version, i.e.,

$$\mathbf{A}_{i,j} := \frac{Z_{i,j}}{\sum_{j=1}^T Z_{i,j}}. \quad (10)$$

We call $\mathbf{A}$ an i.i.d. Markov matrix.

*Remark* 3.2 (Terminology). This is a slight abuse of language since the entries of an "i.i.d. Markov matrix" are not necessarily independent random variables because of the normalisation constraint. Moreover, not all random Markov matrix ensembles satisfy the above definition. Remarkably, the standard key-query dot product attention matrix $\mathbf{A}(\mathbf{X}_0)$ defined in (1) is an i.i.d. Markov matrix provided the input is isotropic $\mathbf{X}_0\mathbf{X}_0^\top = \mathbf{I}$. The uniform attention is another instance from this class of matrices; see Figure 2.

We study as our model a deep stack of attention layers at initialisation, where at each layer $\ell$, the signal is transformed

as $\mathbf{X}_\ell = \mathbf{A}_\ell \mathbf{X}_{\ell-1} \mathbf{W}_\ell^V$. The input signal $\mathbf{X}_0 \in \mathbb{R}^{T \times d}$ has $T$ tokens of embedding dimension $d$, with a fixed ratio $\gamma := \frac{T}{d} \leq 1$. For such a network of *depth* $L$, the input-output relationship is thus given by

$$\mathbf{X}_L = \mathbf{A}_L \mathbf{A}_{L-1} \ldots \mathbf{A}_1 \mathbf{X}_0 \mathbf{W}_1^V \ldots \mathbf{W}_{L-1}^V \mathbf{W}_L^V. \quad (11)$$

The *value matrices* $\mathbf{W}_\ell^V \in \mathbb{R}^{d \times d}$ are initialised independently with i.i.d. $\mathcal{N}(0,1)$ entries and the *attention matrices* $\mathbf{A}_\ell \in \mathbb{R}^{T \times T}$ are independent i.i.d. Markov matrices with $\sigma_A^2 = 1$.

*Remark* 3.3 (Scaling of value matrices). The reason we initialise the value matrices with $\mathcal{N}(0,1)$ entries rather than $\mathcal{N}(0,1/d)$ (i.e. He initialisation) is that the attention matrices have singular values of magnitude $O(1/\sqrt{T})$ except for the leading one $s_1(\mathbf{A}) = 1 + o(1)$; see Theorem A.1. So, in all but one direction, the attention matrix scales down the signal by a factor of $O(1/\sqrt{T})$, which will be compensated by $\mathbf{W}^V$ with singular values of magnitude $O(\sqrt{d})$.

### 3.1. Revisiting rank collapse & exploding gradients

**Rank collapse** Given an isotropic input $\mathbf{X}_0$ with $\mathbf{\Sigma}_0 := \mathbf{X}_0\mathbf{X}_0^\top = \mathbf{I}$, we are interested in understanding how the stable rank of the covariance matrix at layer $\ell$,

$$\mathbf{\Sigma}_\ell := \mathbf{X}_\ell \mathbf{X}_\ell^\top,$$

evolves. Proposition 3.4 demonstrates how the stable rank collapses as the width $T$ increases.

**Proposition 3.4** (Rank collapse in width for deep network). *Assume $\mathbf{\Sigma}_0 = \mathbf{I}$. Then, for any $\ell \geq 1$,*

$$\lim_{T \to \infty} \mathrm{sr}(\mathbf{\Sigma}_\ell) = 1, \quad (12)$$

*with overwhelming probability. Moreover, the convergence happens at a polynomial rate, i.e.*

$$\left| \mathrm{sr}(\mathbf{\Sigma}_\ell) - 1 \right| = O(T^{1-4\ell}).$$

*Remark* 3.5. The proof essentially boils down to determining the gap between the largest and the second largest singular value of $\mathbf{X}_\ell$, since $\mathrm{sr}(\mathbf{\Sigma}_\ell)$ equals

$$\frac{\sum_{i=1}^T s_i^2(\mathbf{\Sigma}_\ell)}{s_1^2(\mathbf{\Sigma}_\ell)} = \frac{\sum_{i=1}^T s_i^4(\mathbf{X}_\ell)}{s_1^4(\mathbf{X}_\ell)} \leq 1 + (T-1)\frac{s_2^4(\mathbf{X}_\ell)}{s_1^4(\mathbf{X}_\ell)}.$$

While $s_1(\mathbf{X}_\ell) = O(1)$ regardless of the layer, the bulk, i.e. $s_2(\mathbf{X}_\ell)$ shrinks like $(1/T)^\ell$, which gives the desired rate.

**Exploding gradients.** A well-known issue that can disrupt training across various neural network architectures is the vanishing or exploding of gradients; see (Hanin, 2018). For attention-only transformers with degenerate attention, (Noci et al., 2022) demonstrate that the gradients with respect to $\mathbf{W}_\ell^V$ vanish. Our model (11) allows for more general

random attention while using a different scaling that makes the same quantity explode rather than vanish. Proposition 3.6 provides a lower bound on the rate at which the gradient grows.

**Proposition 3.6** (Exploding gradients)**.** *For any $L \geq 2$ and $1 \leq \ell \leq L$, with overwhelming probability,*

$$\lim_{T \to \infty} \frac{1}{T^{L-1}} \left\| \frac{\partial \mathbf{X}_L}{\partial \mathbf{W}_\ell^V} \right\|_F^2 \geq C_{L-\ell}, \quad (13)$$

*for some constant $C_{L-\ell} > 0$. In particular, for $T$ large enough, $\|\partial \mathbf{X}_L / \partial \mathbf{W}_\ell^V\|_F^2$ diverges to infinity as $L$ increases. In the single-layer case $\ell = L = 1$, the following improved bound*

$$\lim_{T \to \infty} \frac{1}{T} \left\| \frac{\partial \mathbf{X}_1}{\partial \mathbf{W}_1^V} \right\|_F^2 \geq C \quad (14)$$

*holds almost surely.*

### 3.2. Removing the gap

As previously seen, we can write an i.i.d. Markov matrix $\mathbf{A} \in \mathbb{R}^{T \times T}$ as

$$\mathbf{A} = \frac{1}{T} \mathbf{1}_{T \times T} + \mathbf{A}^\perp, \quad (15)$$

where $\mathbf{1}_{T \times T}$ is the all-ones matrix and $\mathbf{A}^\perp$ has a limiting spectrum resembling that of a Gaussian matrix.

Similar to what we did in the last section to remove the outliers and keep the bulk, we simply replace $\mathbf{A}$ with $\mathbf{A}^\perp$ at every layer, i.e.

$$\mathbf{X}_L^\perp = \mathbf{A}_L^\perp \mathbf{A}_{L-1}^\perp \dots \mathbf{A}_1^\perp \mathbf{X}_0 \mathbf{W}_1^V \dots \mathbf{W}_{L-1}^V \mathbf{W}_L^V, \quad (16)$$

where $\mathbf{A}_\ell^\perp := \mathbf{A}_\ell - T^{-1} \mathbf{1}_{T \times T}$. Note, again, that we directly modify the attention matrices (but *not* the signal representation) and $\mathbf{X}_\ell^\perp$ should be understood as the signal at layer $\ell \geq 1$ in a network whose $\mathbf{A}_\ell$'s are replaced with $\mathbf{A}_\ell^\perp$'s as in (16). We set $\mathbf{X}_0^\perp = \mathbf{X}_0$.

The modified attention has no spectral gap (see Lemma A.6), thus the stable rank of the covariance matrix $\boldsymbol{\Sigma}_\ell^\perp := \mathbf{X}_\ell^\perp \mathbf{X}_\ell^{\perp \top}$ does not collapse anymore, as detailed in Proposition 3.7 (cf. Proposition 3.4).

**Proposition 3.7** (Resolved rank collapse in width)**.** *Let $\mathbf{X}_\ell^\perp = \mathbf{A}_\ell^\perp \mathbf{X}_{\ell-1}^\perp \mathbf{W}_\ell^V$ be the signal at layer $\ell$ in our modified model (16) and $\boldsymbol{\Sigma}_\ell^\perp \in \mathbb{R}^{T \times T}$ be its covariance matrix. Then, almost surely, the rank does not collapse, i.e., there exists a constant $C_\ell > 0$ such that,*

$$\lim_{T \to \infty} \frac{\mathrm{sr}(\boldsymbol{\Sigma}_\ell^\perp)}{T} = C_\ell. \quad (17)$$

Our modification also mitigates the average growth of the gradients. Proposition 3.8 establishes a linear growth rate for $\|\partial \mathbf{X}_L^\perp / \partial \mathbf{W}_\ell^V\|_F^2$ in expectation, which should be compared to the rate of $T^{L-1}$ from Proposition 3.6.

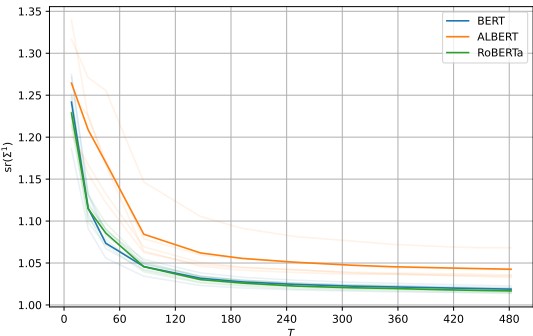

(a) Rank collapse in width

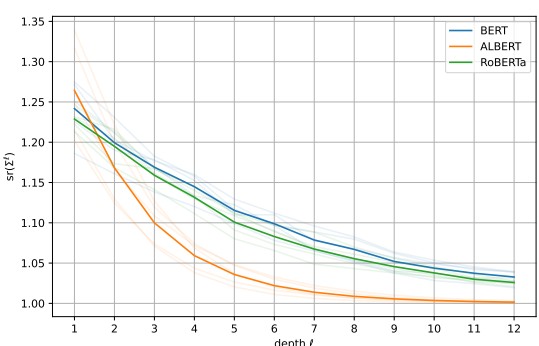

(b) Rank collapse in depth

*Figure 3.* Famous transformer encoders suffer from rank collapse at initialisation, both in (a) width and (b) depth. These untrained models ($d = 784$) are loaded from Hugging Face and sentences from this paper's abstract are tokenised using pre-trained tokenisers. Inputs are therefore non-isometric (see Appendix A.1 for a more detailed discussion).

**Proposition 3.8** (Resolved exploding gradients)**.** *Let $\mathbf{X}_\ell^\perp = \mathbf{A}_\ell^\perp \mathbf{X}_{\ell-1}^\perp \mathbf{W}_\ell^V$ be the signal at layer $\ell$ in our modified model (16). Then, in expectation, the squared norm of the gradients grow linearly with $d$, i.e. there exists a constant $C > 0$ such that,*

$$\lim_{d \to \infty} \frac{1}{d} \mathbb{E} \left\| \frac{\partial \mathbf{X}_L^\perp}{\partial \mathbf{W}_\ell^V} \right\|_F^2 = C. \quad (18)$$

## 4. Experiments and Further Insights

**Rank collapse observed in practice.** We highlight the practical relevance of our findings by showing rank collapse occurs both in width and depth in widely used transformer encoders like BERT, see Figure 3. In Figure 4(a), all solid lines, corresponding to the vanilla attention layer block, show a stable rank that rapidly converges to 1 as the context length $T$ grows (at a rate given in Theorem 2.1). This occurs even if additional components such as LayerNorm,

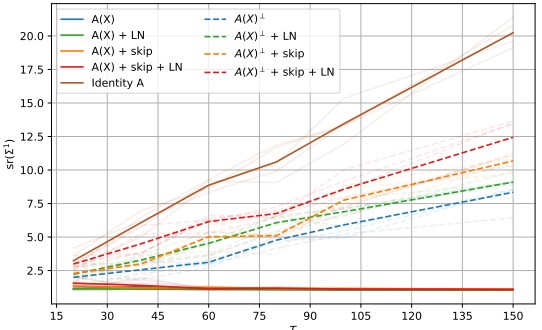

(a) Spectral gap causes rank collapse in width.

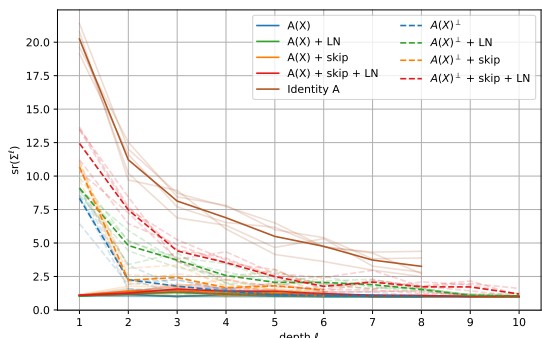

(b) The rank inevitably collapses in depth.

*Figure 4.* Rank collapse occurs both in width and depth. Our fix effectively prevents the rank from collapsing in width within an attention layer. Although rank collapse in depth occurs regardless of the presence of the spectral gap, our fix consistently slows the collapse—a feat no other module achieves.

skip connections, or both are added on top of the attention block. Provided that rank collapse in depth is an inherent consequence of repeated matrix multiplications, architectural modifications of the attention layer can only slow the collapse rather than completely prevent it. We demonstrate this by showing that rank collapse in depth persists even when the attention matrix is set to the identity matrix—an extreme case with the highest possible stable rank and no spectral gap.

**Removing the spectral gap.**  As an input signal propagates through an initialised transformer, we can address both forms of rank collapse—across width and depth—by eliminating the spectral gap induced by the attention matrix at each layer. Figure 4(a) reinforces our findings, showing that our solution—indicated by $\perp$ in the legend—consistently mitigates rank collapse. Remarkably, the fix is effective even in deep transformers when combined with additional components such as LayerNorm, skip connections, or both.

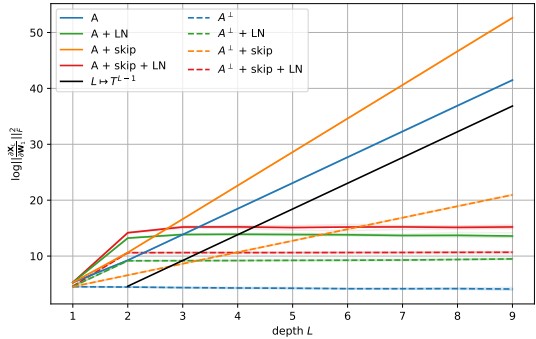

*Figure 5.* In deep transformers initialised with i.i.d. Markov attention, our "remove the gap" fix is effective, as we can precisely discard the single outlier in the spectrum.

Indeed, the stable rank now scales up with the context length $T$ (at a linear rate predicted by our theory), in contrast to the original behaviour where it would quickly vanish to $1$. Once the rank collapse in width is mitigated, it naturally slows down the one in depth, as seen in Figure 4(b). Another possible way to slow down rank collapse in depth (though not eliminate it) is to set the value matrices as orthogonal matrices.

After passing an isotropic input through the network, we compute the gradient norm as defined in (13). When the attention is an i.i.d. Markov matrix, gradient norms are effectively controlled with depth by either applying Layer-Norm or removing the spectral gap, as illustrated in Figure 5. The corresponding lines remain flat, confirming the theoretically expected constant trend with $L$. Without these solutions, the gradients explode exponentially with depth as indicated by the linear trend in the log-log scale, with the derived lower bound of $T^{L-1}$ empirically confirmed.

**Future avenues.**  Beyond one attention layer, we conduct an empirical investigation of the spectrum of the actual attention matrix across layers. Surprisingly, as shown in Figure 4, we observe the emergence of additional outliers in the spectrum. We assess the effectiveness of our proposed solution—removing the largest spectral outlier—by comparing it to an alternative approach, which consists in removing all outliers. Since identifying and locating these additional outliers is highly non-trivial, we perform an SVD of the attention matrix at each layer and manually identify and remove the extra gaps based on the largest difference between singular values. As depicted in Figure 6, this computationally expensive approach does not significantly outperform our proposed fix, which is much more efficient. In fact, the two methods achieve comparable results in mitigating rank collapse (first in width, then in depth) and controlling

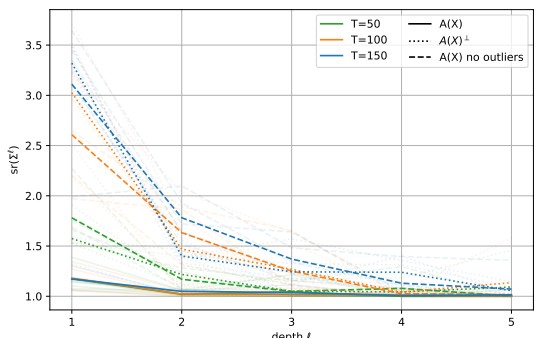

*Figure 6.* In deep transformers, our "remove the gap" fix is nearly as effective as removing all outliers from the spectrum despite having a significantly lower computational cost.

gradient norms. This demonstrates that removing the largest spectral gaps yields the most significant benefit with the lowest cost. Finally, while training dynamics are beyond the scope of this work, our theoretical framework provides a fresh perspective on recent empirical studies that report significant training benefits from various ad hoc fixes. Our analysis offers insights into *why* these fixes may be effective. These studies include approaches such as "centering" self-attention layers (Ali et al., 2023), where the authors remove exactly $T^{-1}\mathbf{1}_{T \times T}$ from the attention layer, and "differentiating" the attention mechanism, as seen in the Differential Transformer architecture, where two softmax attention matrices are subtracted from each other at each layer to "cancel the noise"; see (Ye et al., 2025). We can now interpret these approaches as implicit efforts to address the spectral gap.

## 5. Conclusion

We introduced a new mathematical framework for studying the self-attention mechanism at initialisation, leveraging results from random matrix theory and free probability. By analysing the spectral properties of i.i.d. Markov matrices, we diagnosed random softmax-based attention with a spectral gap that leads to rank collapse in width—a phenomenon revealed and demonstrated for the first time by our analysis—alongside the previously established rank collapse in depth and exploding gradients.

We proposed a simple modification of the attention mechanism, which is provably effective in slowing rank collapse. Additionally, we empirically discovered that the spectra of standard key-query attention matrices often feature multiple outliers in deeper layers. Context length appears to have been overlooked in previous analyses of rank collapse in favour of depth. Given the continuous growth in the size of transformer models used in practice, we hope this work encourages the community to reconsider the emphasis on depth in theoretical developments, particularly in the analysis of rank collapse.

## Acknowledgements

TNS is supported by the UK Engineering and Physical Sciences Research Council through the grant EP/W523781/1. JT acknowledges support from His Majesty's Government in the development of this research as well as by EPSRC grant EP/Y028872/1, Mathematical Foundations of Intelligence: An "Erlangen Programme" for AI.

## Impact Statement

This paper presents work whose goal is to advance the field of Machine Learning. There are many potential societal consequences of our work, none which we feel must be specifically highlighted here.

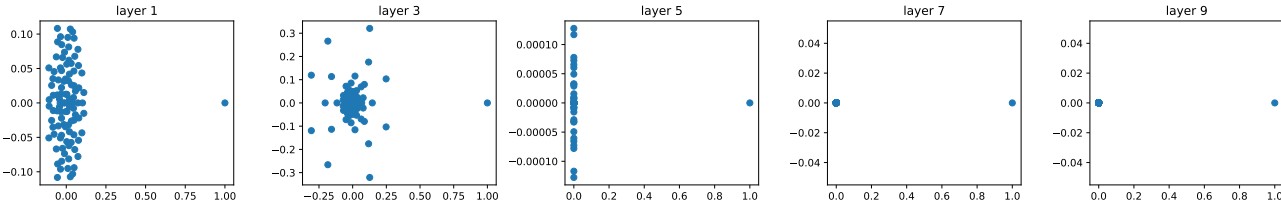

*Figure 7.* Spectrum of a $T \times T$ softmax-based attention matrix (with $T = 200$) across layers. Some additional outlier eigenvalues emerge from the spectrum with depth in an intricate way. The matrix, however, consistently converges to uniform attention for large $\ell$.

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

# A. Appendix

The Appendix is organised as follows:

- Section A.1 provides strong empirical support for the main theorem of this paper even when its assumptions are violated in practice.

- Section A.2 presents the proofs of the paper's propositions.

- Section A.3 provides auxiliary lemmas along with their proofs that support the previous section.

- Section A.4 explores dynamical isometry in transformers, questioning their ability to propagate information *faithfully* through attention layers once the problematic outliers have been removed.

- Section A.5 outlines comprehensive experimental details, ensuring reproducibility.

## A.1. Justification of our theoretical assumptions

In this section, we empirically support the finding of our main theorem—a spectral gap arises in width within attention layers—even when its assumptions are violated. Our formal proof should be seen more as a proof of concept, and its conditions to hold are not necessary in practice, but rather made for mathematical rigor.

**The inputs do not need to be isometric in practice.** In Figure 8, sentences are tokenised with a pre-trained tokeniser, making inputs to our randomly initialised BERT clearly non-isometric. We include $\mathbf{X}_0\mathbf{X}_0^T$ to illustrate its deviation from identity. When tokens are instead assigned random embeddings as in Figure 9, the covariance matrix is closer to orthogonality. Yet, the spectral gap and rank collapse persist in both cases. We also show the spectra of a random attention layer with (synthetic) non-isometric input tokens to mirror Figure 1 of the main in Figure 10.

**The context length $T$ does not need to be bigger than the token dimension $d$ in practice.** Note that without $T \leq d$, the inputs cannot be isometric so this extra assumption was solely made for this reason. We provide the reader with an ablation on the role of $\gamma$, the ratio between the context length and the token dimension. In Figure 10, we explore information propagation in TinyBert, for which $\gamma \geq 1$, hence necessarily the tokens must be non-orthonormal. The spectral gap persists, and rank collapse in width follows. Another ablation, this time conducted on synthetic data, can be found in Figure 10.

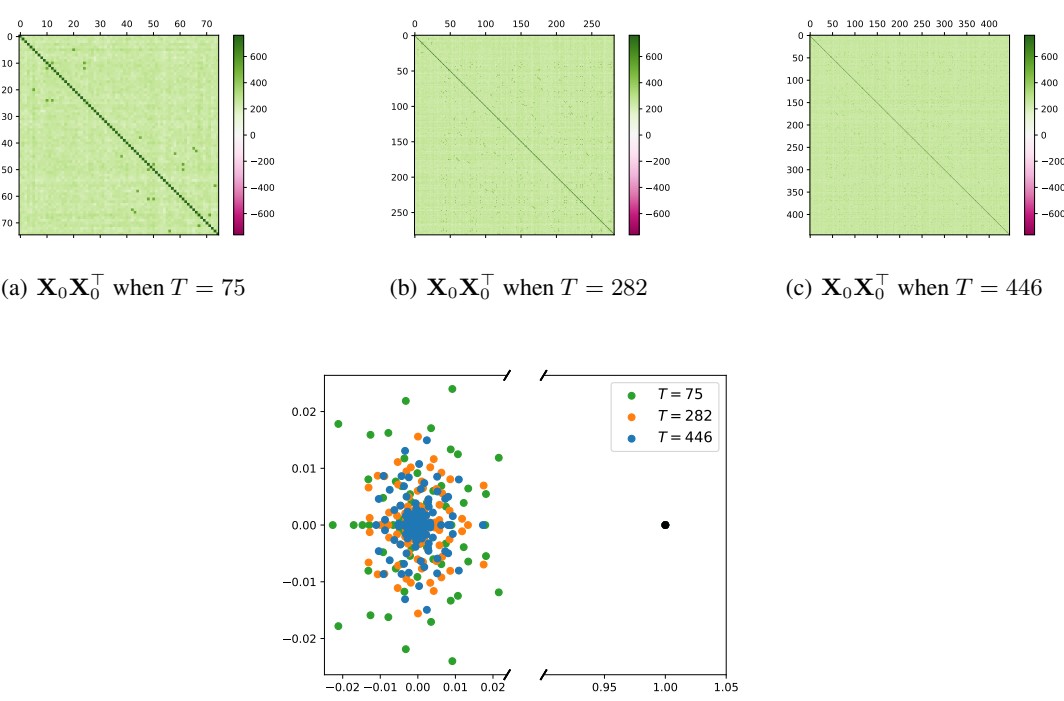

(a) $\mathbf{X}_0\mathbf{X}_0^\top$ when $T = 75$

(b) $\mathbf{X}_0\mathbf{X}_0^\top$ when $T = 282$

(c) $\mathbf{X}_0\mathbf{X}_0^\top$ when $T = 446$

(d) Eigenvalues in the complex plane of the attention matrix in the first head of layer 1.

*Figure 8.* Spectral gap (second row) persists even when the inputs are not orthogonal (see first row). Experiment conducted with a randomly initialised RoBERTa encoder, hence $d = 768$ and $T < 512$ by construction, on real text data from our abstract, processed by a pretrained tokenizer available on HuggingFace. We present a single realisation, though this behavior is consistently observed across multiple runs.

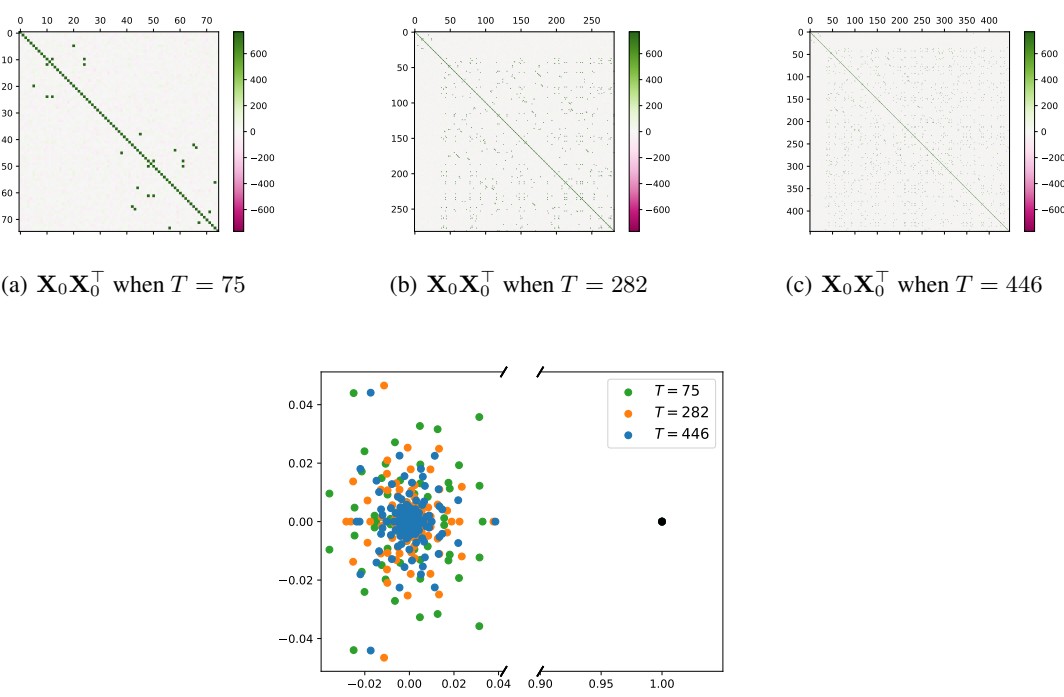

(a) $\mathbf{X}_0\mathbf{X}_0^\top$ when $T = 75$

(b) $\mathbf{X}_0\mathbf{X}_0^\top$ when $T = 282$

(c) $\mathbf{X}_0\mathbf{X}_0^\top$ when $T = 446$

(d) The eigenvalues in the complex plane formed by the attention matrix in the first head of layer 1 align with our proof of concept.

*Figure 9.* Spectral gap (second row) persists on non-synthetic yet random data. Using a randomly initialized RoBERTa encoder ($d = 768$, $T < 512$ by construction), we substitute the word embeddings given by the pretrained tokenizer with random ones, making the input covariance matrix closer to identity (first row). We present a single realisation, though this behavior is consistently observed across multiple runs.

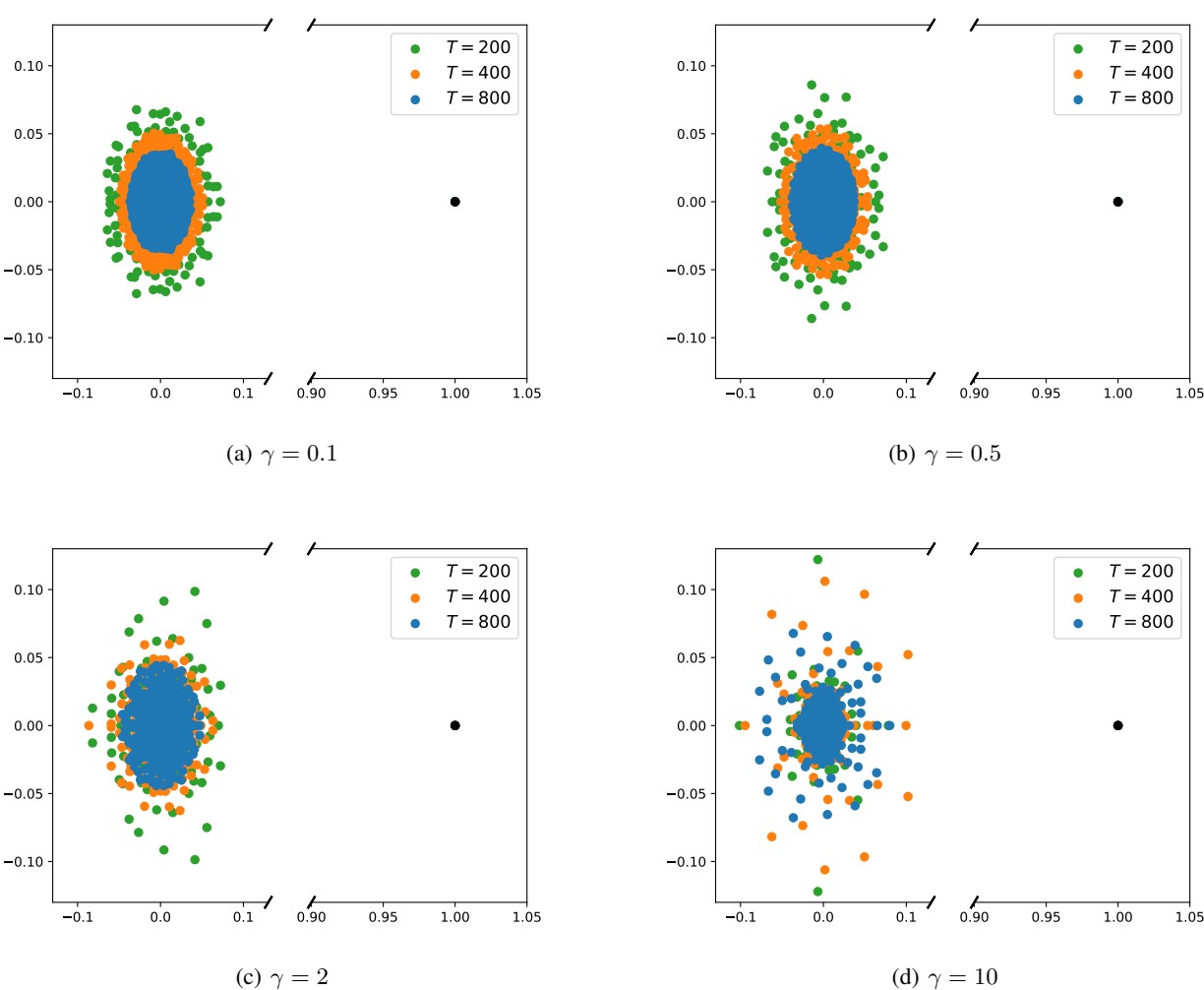

*Figure 10.* The eigenvalue distribution of a single attention layer for non-isometric synthetic input and different values of $\gamma = \frac{T}{d}$. The input is generated by drawing i.i.d. Gaussian entries followed by normalising the rows, so each token has unit length but they are not necessarily orthogonal. The gap between the bulk and the edge persists despite the bulk looking different for $\gamma > 1$.

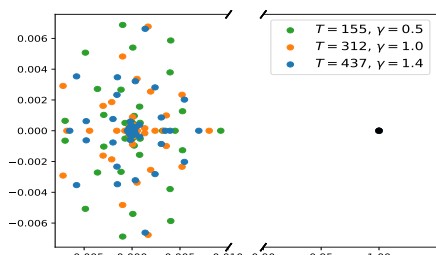

(a) Spectral gap in the attention matrix of the first head of layer 1.

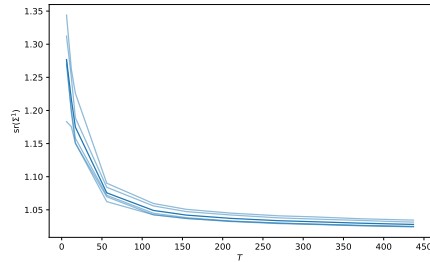

(b) Stable rank quickly collapses in width, even when $\gamma > 1$.

*Figure 11.* Spectral gap (top) persists in TinyBert encoders for which $\gamma > 1$, highlighting the generality of our result, which constrained the value of $\gamma$ only for the sake of having a mathematical proof. As an almost direct consequence of the spectral gap in the attention matrix, stable rank rapidly collapses (bottom) in width. Here $d = 312$ by design and we increase $\gamma$ by feeding the network with longer sequences. Whilst the eigenvalues (top) are shown for one specific realisation (but consistently observed across runs), the stable rank (bottom) is averaged over 5 runs.

## A.2. Proofs

*Proof of Theorem 2.1.* Let us show that the attention matrix $\mathbf{A}(\mathbf{X}_0)$ satisfies Definition 3.1 by demonstrating that the random variables

$$Z_{i,j} := \exp\left(\frac{\mathbf{X}_0 \mathbf{W}_1^Q \mathbf{W}_1^{K^\top} \mathbf{X}_0^\top}{\sqrt{d_{qk}}}\right)_{i,j}$$

are i.i.d. with a finite fourth moment.

Since the key and query matrices are initialised as Gaussian i.i.d. matrices and the input data $\mathbf{X}_0$ is isotropic, $\widetilde{\mathbf{W}}^Q := \mathbf{X}_0 \mathbf{W}_1^Q$ and $\widetilde{\mathbf{W}}^K := \mathbf{X}_0 \mathbf{W}_1^K$ follow the same distribution as $\mathbf{W}_1^Q$ and $\mathbf{W}_1^K$. Each $Z_{i,j}$ can be written as the exponential of the inner product between the $i$-th row of $\widetilde{\mathbf{W}}^Q$ and the $j$-th row of $\widetilde{\mathbf{W}}^K$, thus $Z_{i,j}$ are i.i.d. and we only need to prove that $\mathbb{E}|Z_{1,1}|^4 < \infty$. Let us define

$$U_{d_{qk}} := \sum_{r=1}^{d_{qk}} \widetilde{\mathbf{W}}_{1,r}^Q \widetilde{\mathbf{W}}_{1,r}^K$$

to be the dot product of the first row of $\widetilde{\mathbf{W}}^Q$ and the first row of $\widetilde{\mathbf{W}}^K$. So, $U_{d_{qk}}$ is simply the sum of $d_{qk}$ i.i.d. copies of $U_1$, the product of two independent Gaussian random variables, whose density is known to be

$$f_1(x) := \frac{1}{\pi \sigma_{qk}^2} K_0\left(\frac{|x|}{\sigma_{qk}^2}\right),$$

where $K_0$ is the modified Bessel function of the second kind. Therefore, the probability density function of $U_{d_{qk}}$ is given by the $d_{qk}$-fold convolution

$$f_{d_{qk}}(x) = \underbrace{f_1(x) * \cdots * f_1(x)}_{d_{qk} \text{ times}}.$$

It is also known that $K_0(x)$ asymptotically behaves like $\sqrt{\frac{\pi}{2x}} e^{-x}$ and that the convolution $g * h$ decays at least as fast as the slower of $g$ and $h$. Combining these facts, we conclude that $f_{d_{qk}}$ decays at least as fast as $e^{-x}$, i.e.

$$f_{d_{qk}}(x) = g(|x|) e^{-|x|},$$

for some polynomially-bounded $g$. Now we can bound our quantity of interest

$$\begin{aligned}
\mathbb{E}|Z_{1,1}|^4 &= \mathbb{E}\left[\exp\left(\frac{4U_{d_{qk}}}{\sqrt{d_{qk}}}\right)\right] \\
&= \int_{\mathbb{R}} e^{4x/\sqrt{d_{qk}}} \, g(|x|) e^{-|x|} dx \\
&< \infty,
\end{aligned}$$

as long as $\frac{4}{\sqrt{d_{qk}}} < 1$, i.e. $d_{qk} > 16$.

It is now time to invoke an important result, that we first state.

**Theorem A.1** (Spectral gap in i.i.d. Markov matrices, (Bordenave et al., 2011)). *Let $\mathbf{A} \in \mathbb{R}^{T \times T}$ be an i.i.d. Markov matrix whose underlying distribution has variance $\sigma_A^2$. Then, $\lambda_1(\mathbf{A}) = 1$ and almost surely,*

$$\lim_{T \to \infty} s_1(\mathbf{A}) = 1 \quad \text{and} \quad \lim_{T \to \infty} s_2(\sqrt{T}\mathbf{A}) = 2\sigma_A \quad \text{while} \quad \overline{\lim}_{T \to \infty} |\lambda_2(\sqrt{T}\mathbf{A})| \leq 2\sigma_A. \tag{19}$$

Having shown that $\mathbf{A}(\mathbf{X}_0)$ forms an i.i.d. Markov matrix since the $Z_{i,j}$'s are i.i.d. with finite fourth moment, the spectral gap as described in Theorem 2.1 follows immediately. $\qquad\square$

*Proof of Proposition 3.4.* Fix $\ell \geq 1$. By definition of stable rank, we have

$$\begin{aligned}
\mathrm{sr}(\mathbf{\Sigma}_\ell) &= \frac{\sum_{i=1}^T s_i^2(\mathbf{\Sigma}_\ell)}{s_1^2(\mathbf{\Sigma}_\ell)} = \frac{\sum_{i=1}^T s_i^2(\mathbf{X}_\ell \mathbf{X}_\ell^\top)}{s_1^2(\mathbf{X}_\ell \mathbf{X}_\ell^\top)} = \frac{\sum_{i=1}^T s_i^4(\mathbf{X}_\ell)}{s_1^4(\mathbf{X}_\ell)} \\
&= 1 + \sum_{i=2}^T \frac{s_i^4(\mathbf{X}_\ell)}{s_1^4(\mathbf{X}_\ell)} \leq 1 + (T-1)\frac{s_2^4(\mathbf{X}_\ell)}{s_1^4(\mathbf{X}_\ell)}.
\end{aligned}$$

For $T$ large enough, let us say bigger than some $T_0$, Theorem A.5 provides a deterministic upper bound, i.e. $s_2(\mathbf{X}_\ell) \leq K$ for some constant $K$. Moreover, Theorem A.4 gives the bound $T^{-\ell}s_1(\mathbf{X}_\ell) \in (1-t, 1+t)$ with (overwhelming) probability $P_{t,T}$ for arbitrary $t > 0$ and $T$ bigger than some $T_1$. Thus, for $T \geq \max(T_0, T_1)$,

$$1 \leq \mathrm{sr}(\mathbf{\Sigma}_\ell) \leq 1 + \frac{(T-1)K^4}{T^{4\ell}(1-t)^4}$$

with probability at least $P_{t,T}$. Therefore, the event

$$\lim_{T \to \infty} \mathrm{sr}(\mathbf{\Sigma}_\ell) = 1$$

holds with overwhelming probability. $\qquad \square$

*Proof of Proposition 3.6.* Note that we will treat the matrix-to-matrix derivatives such as $\partial\mathbf{X}_L/\partial\mathbf{W}_\ell^V$ not as a tensor (in $\mathbb{R}^{T \times d \times d \times d}$), but as its matricised version (in $\mathbb{R}^{Td \times d^2}$). We make use of the chain rule to compute the gradients of interest. Namely, at layer $\ell$,

$$
\begin{aligned}
\frac{\partial\mathbf{X}_L}{\partial\mathbf{W}_\ell^V} &= \frac{\partial\mathbf{X}_L}{\partial\mathbf{X}_\ell}\frac{\partial\mathbf{X}_\ell}{\partial\mathbf{W}_\ell^V} \\
&= \Big((\mathbf{A}_L \dots \mathbf{A}_{\ell+1}) \otimes (\mathbf{W}_{\ell+1}^V \dots \mathbf{W}_L^V)\Big)\Big((\mathbf{A}_\ell \dots \mathbf{A}_1\mathbf{X}_0\mathbf{W}_1^V \dots \mathbf{W}_{\ell-1}^V) \otimes \mathbf{I}_d\Big) \\
&= \big(\underbrace{\mathbf{A}_L \dots \mathbf{A}_1\mathbf{X}_0\mathbf{W}_1^V \dots \mathbf{W}_{\ell-1}^V}_{:=\mathbf{P}_1}\big) \otimes \big(\underbrace{\mathbf{W}_{\ell+1}^V \dots \mathbf{W}_L^V}_{:=\mathbf{P}_2}\big).
\end{aligned}
$$

Then, by properties of Kronecker product, we have

$$\left\|\frac{\partial\mathbf{X}_L}{\partial\mathbf{W}_\ell^V}\right\|_F^2 = \sum_i s_i^2\Big(\frac{\partial\mathbf{X}_L}{\partial\mathbf{W}_\ell^V}\Big) = \sum_{i,j} s_i^2(\mathbf{P}_1)s_j^2(\mathbf{P}_2) \geq s_1^2(\mathbf{P}_1)s_1^2(\mathbf{P}_2). \tag{20}$$

The largest singular value of a product of i.i.d. Gaussian matrices has been studied extensively, e.g., see (Akemann et al., 2013; Młotkowski et al., 2015). (Nait Saada & Naderi, 2024) show that, almost surely,

$$s_1^2(\mathbf{P}_2) = T^{L-\ell}\frac{(L-\ell+1)^{L-\ell+1}}{(L-\ell)^{L-\ell}}(1 + o(1)).$$

On the other hand, by Theorem A.4, $s_1(\mathbf{P}_1)$ concentrates around $T^{\frac{\ell-1}{2}}$ with overwhelming probability, i.e., for $T$ large enough,

$$s_1^2(\mathbf{P}_1) \in (T^{\ell-1}(1-t)^2, T^{\ell-1}(1+t)^2),$$

with probability at least $P_{t,T}$. Altogether, with an overwhelming probability we have

$$\lim_{T \to \infty} \frac{1}{T^{L-1}}\left\|\frac{\partial\mathbf{X}_L}{\partial\mathbf{W}_\ell^V}\right\|_F^2 \geq (1-t)^2\frac{(L-\ell+1)^{L-\ell+1}}{(L-\ell)^{L-\ell}}.$$

One can get a better bound in the single-layer case ($\ell = L = 1$). Since $\mathbf{P}_2 = \mathbf{I}_d$, we can rewrite (20) as

$$\left\|\frac{\partial\mathbf{X}_1}{\partial\mathbf{W}_1^V}\right\|_F^2 = \sum_i s_i^2\Big(\frac{\partial\mathbf{X}_1}{\partial\mathbf{W}_1^V}\Big) = \sum_i s_i^2(\mathbf{P}_1)\sum_j s_j^2(\mathbf{I}_d) \geq d \cdot s_1^2(\mathbf{P}_1),$$

while $s_1^2(\mathbf{P}_1) = s_1^2\big(\mathbf{A}_1(\mathbf{X}_0)\mathbf{X}_0\big) = O(1)$ almost surely. Therefore, almost surely, the following improved bound

$$\lim_{T \to \infty} \frac{1}{T}\left\|\frac{\partial\mathbf{X}_1}{\partial\mathbf{W}_1^V}\right\|_F^2 \geq C,$$

holds. $\qquad \square$

*Proof of Proposition 3.7.* The resolved stable rank can be written as,

$$\frac{\mathrm{sr}(\mathbf{\Sigma}_\ell^\perp)}{T} = \frac{T^{-1}\sum_{i=1}^T s_i^2(\mathbf{A}_\ell^\perp \dots \mathbf{A}_1^\perp \mathbf{X}_0 \mathbf{W}_1^V \dots \mathbf{W}_\ell^V)}{s_1^2(\mathbf{A}_\ell^\perp \dots \mathbf{A}_1^\perp \mathbf{X}_0 \mathbf{W}_1^V \dots \mathbf{W}_\ell^V)}.$$

By submultiplicativity of the operator norm,

$$\frac{\mathrm{sr}(\mathbf{\Sigma}_\ell^\perp)}{T} \geq \frac{T^{-1}\sum_{i=1}^T s_i^2(\mathbf{A}_\ell^\perp \dots \mathbf{A}_1^\perp \mathbf{X}_0 \mathbf{W}_1^V \dots \mathbf{W}_\ell^V)}{s_1^2(\mathbf{A}_\ell^\perp) \dots s_1^2(\mathbf{A}_1^\perp) s_1^2(\mathbf{X}_0 \mathbf{W}_1^V) \dots s_1^2(\mathbf{W}_\ell^V)}.$$

Let us call $P_T$ the fraction of squared singular values of $\mathbf{A}_\ell^\perp \dots \mathbf{A}_1^\perp \mathbf{X}_0 \mathbf{W}_1^V \dots \mathbf{W}_\ell^V$ above a certain finite threshold $c$, i.e.

$$P_T := \frac{1}{T}\sum_{i=1}^T \mathbb{1}_{s_i^2(\mathbf{A}_\ell^\perp \dots \mathbf{A}_1^\perp \mathbf{X}_0 \mathbf{W}_1^V \dots \mathbf{W}_\ell^V) > c}.$$

Then, trivially

$$\frac{\mathrm{sr}(\mathbf{\Sigma}_\ell^\perp)}{T} \geq \frac{c\, P_T}{s_1^2(\mathbf{A}_\ell^\perp) \dots s_1^2(\mathbf{A}_1^\perp) s_1^2(\mathbf{X}_0 \mathbf{W}_1^V) \dots s_1^2(\mathbf{W}_\ell^V)}. \tag{21}$$

Assuming the asymptotic freeness of all attention matrices $\mathbf{A}_1^\perp, \dots, \mathbf{A}_\ell^\perp$ and weight matrices $\widetilde{\mathbf{W}}_1^V = \mathbf{X}_0 \mathbf{W}_1, \mathbf{W}_2, \dots, \mathbf{W}_\ell$, we may write the limiting squared singular value distribution of $\mathbf{X}_\ell^\perp$ as the free convolution of the corresponding Marchenko-Pastur distributions:

$$\mathcal{M} := \mathcal{MP}^{\boxtimes \ell}(1, \sigma_A) \boxtimes \mathcal{MP}(\gamma, \frac{\sigma_W}{\sqrt{\gamma}}) \boxtimes \mathcal{MP}^{\boxtimes \ell-1}(1, \frac{\sigma_W}{\sqrt{\gamma}}).$$

Then, almost surely,

$$P_T \longrightarrow P := \int_c^\infty d\mathcal{M}.$$

The distribution $\mathcal{M}$ is compactly supported on the interval $[0, s_{\gamma,2\ell}^+]$, where $s_{\gamma,2\ell}^+$ does not depend on $T$. So, by choosing $c < s_{\gamma,2\ell}^+$, we can make $c\,P$ a non-zero constant. Moreover, the denominator of (21) converges almost surely to some constant (in $T$), i.e.

$$s_1^2(\mathbf{A}_\ell^\perp) \dots s_1^2(\mathbf{A}_1^\perp) s_1^2(\mathbf{X}_0 \mathbf{W}_1^V) \dots s_1^2(\mathbf{W}_\ell^V) \to (2\sigma_A)^{2\ell}\sigma_W^{2\ell}2^{2(\ell-1)}(1 + \gamma^{-1/2})^2.$$

Thus, almost surely,

$$\lim_{T\to\infty} \frac{\mathrm{sr}(\mathbf{\Sigma}_\ell^\perp)}{T} \geq \frac{c\,P}{(\sigma_A\sigma_W)^{2\ell}4^{\ell-1}(1 + \gamma^{-1/2})^2} > 0.$$

$\square$

*Proof of Proposition 3.8.* Let us compute the resolved gradients:

$$\frac{\partial \mathbf{X}_L^\perp}{\partial \mathbf{W}_\ell^V} = \frac{\partial \mathbf{X}_L^\perp}{\partial \mathbf{X}_\ell^\perp}\frac{\partial \mathbf{X}_\ell^\perp}{\partial \mathbf{W}_\ell^V}$$

$$= \big(\underbrace{\mathbf{A}_L^\perp \dots \mathbf{A}_1^\perp \mathbf{X}_0 \mathbf{W}_1^V \dots \mathbf{W}_{\ell-1}^V}_{:=\mathbf{P}_1^\perp}\big) \otimes \big(\underbrace{\mathbf{W}_{\ell+1}^V \dots \mathbf{W}_L^V}_{=\mathbf{P}_2}\big).$$

Therefore,

$$\mathbb{E}\left\|\frac{\partial \mathbf{X}_L^\perp}{\partial \mathbf{W}_\ell^V}\right\|_F^2 = \mathbb{E}\Big[\mathrm{tr}\Big(\frac{\partial \mathbf{X}_L^\perp}{\partial \mathbf{W}_\ell^V}\big(\frac{\partial \mathbf{X}_L^\perp}{\partial \mathbf{W}_\ell^V}\big)^\top\Big)\Big]$$

$$= \mathbb{E}\Big[\mathrm{tr}\big(\mathbf{P}_1^\perp(\mathbf{P}_1^\perp)^\top\big)\,\mathrm{tr}\big(\mathbf{P}_2\mathbf{P}_2^\top\big)\Big].$$

Assuming $\mathbf{P}_1^{\perp}$ and $\mathbf{P}_2$ are asymptotically free, we have

$$\lim_{d\to\infty} \frac{1}{d^2} \mathbb{E}\left\|\frac{\partial \mathbf{X}_L^{\perp}}{\partial \mathbf{W}_{\ell}^V}\right\|_F^2 = \lim_{d\to\infty} \frac{1}{d}\mathbb{E}\left(\operatorname{tr}\left(\mathbf{P}_1^{\perp}(\mathbf{P}_1^{\perp})^{\top}\right)\right) \lim_{d\to\infty} \frac{1}{d}\mathbb{E}\left(\operatorname{tr}\left(\mathbf{P}_2\mathbf{P}_2^{\top}\right)\right).$$

For each product matrix, the normalised expectation on the RHS of the above converges to the first moment of its limiting squared singular value distribution. By scaling them properly, i.e.

$$\widetilde{\mathbf{P}}_1^{\perp} := \sqrt{T}\mathbf{A}_L^{\perp} \ldots \sqrt{T}\mathbf{A}_1^{\perp} \frac{1}{\sqrt{d}}\mathbf{X}_0 \mathbf{W}_1^V \ldots \frac{1}{\sqrt{d}}\mathbf{W}_{\ell-1}^V = T^{L/2}d^{-(\ell-1)/2}\mathbf{P}_1^{\perp},$$

$$\widetilde{\mathbf{P}}_2 := \frac{1}{\sqrt{d}}\mathbf{W}_{\ell+1}^V \ldots \frac{1}{\sqrt{d}}\mathbf{W}_L^V = d^{-(L-\ell)/2}\mathbf{P}_2,$$

we make sure that those limiting distributions (free convolutions of Marchenko-Pastur distributions) are compactly supported on an interval of length $O(1)$ and, hence, both $C_1 := \lim \mathbb{E}(\operatorname{tr}(\widetilde{\mathbf{P}}_1^{\perp}(\widetilde{\mathbf{P}}_1^{\perp})^{\top}))$ and $C_2 := \lim \mathbb{E}(\operatorname{tr}(\widetilde{\mathbf{P}}_2(\widetilde{\mathbf{P}}_2)^{\top}))$ are constants. Thus, since $T = \gamma d$,

$$\lim_{d\to\infty} \frac{1}{d^2} \mathbb{E}\left\|\frac{\partial \mathbf{X}_L^{\perp}}{\partial \mathbf{W}_{\ell}^V}\right\|_F^2 = C_1(T^{-L}d^{\ell-1}) \cdot C_2(d^{L-\ell}) = Cd^{-1},$$

or

$$\lim_{d\to\infty} \frac{1}{d} \mathbb{E}\left\|\frac{\partial \mathbf{X}_L^{\perp}}{\partial \mathbf{W}_{\ell}^V}\right\|_F^2 = C.$$

$\square$

*Proof of Theorem A.8.* Since $T = \gamma d$, we can write

$$\mathbf{X}_{\ell} = \mathbf{A}_{\ell}^{\perp} \ldots \mathbf{A}_1^{\perp}\mathbf{X}_0\mathbf{W}_1^V \ldots \mathbf{W}_{\ell}^V$$
$$= \sqrt{T}\mathbf{A}_{\ell}^{\perp} \ldots \sqrt{T}\mathbf{A}_1^{\perp} \frac{1}{\sqrt{d}}\left(\frac{1}{\sqrt{\gamma}}\mathbf{X}_0\mathbf{W}_1^V\right) \ldots \frac{1}{\sqrt{d}}\left(\frac{1}{\sqrt{\gamma}}\mathbf{W}_{\ell}^V\right).$$

Each of the rescaled matrices above has squared singular values that almost surely follow a Marchenko-Pastur distribution $\mathcal{MP}(p, \alpha)$, where $p$ is the ratio between the numbers of rows and columns of each matrix, and $\alpha$ the variance of its entries. Therefore, almost surely, the squared singular values of $\mathbf{X}_{\ell}$, or equivalently the singular values of $\mathbf{\Sigma}_{\ell}$, follow a distribution $\mathcal{M}$ which is given by the free convolution

$$\mathcal{M} := \mathcal{MP}^{\boxtimes\ell}(1, \sigma_A) \boxtimes \mathcal{MP}(\gamma, \sigma_V/\sqrt{\gamma}) \boxtimes \mathcal{MP}^{\boxtimes\ell-1}(1, \sigma_V/\sqrt{\gamma}).$$

The moments of such a distribution are given by Lemma A.7 in the general case. Substituting the corresponding values from our setting gives the desired result.

$\square$

*Proof of Theorem A.9.* Let $\mathbf{A}^{\perp} := \mathbf{A}_{\ell}^{\perp} \ldots \mathbf{A}_1^{\perp} \in \mathbb{R}^{T\times T}$ and $\mathbf{W}^V := \mathbf{W}_1^V \ldots \mathbf{W}_{\ell}^V \in \mathbb{R}^{d\times d}$. Then

$$\mathbf{J}_{\ell} = \mathbf{A}^{\perp} \otimes \mathbf{W}^V \in \mathbb{R}^{Td\times Td},$$

and we can compute the $k$-th moment of its limiting squared singular value distribution as

$$\lim_{T,d\to\infty} \mathbb{E}\left[\frac{1}{Td}\operatorname{tr}(\mathbf{J}_{\ell}\mathbf{J}_{\ell}^{\top})^k\right] = \lim_{T,d\to\infty} \mathbb{E}\left[\frac{1}{Td}\operatorname{tr}\left((\mathbf{A}^{\perp}\mathbf{A}^{\perp\top} \otimes \mathbf{W}^V\mathbf{W}^{V\top})^k\right)\right]$$
$$= \lim_{T,d\to\infty} \mathbb{E}\left[\frac{1}{T}\operatorname{tr}\left((\mathbf{A}^{\perp}\mathbf{A}^{\perp\top})^k\right)\frac{1}{d}\operatorname{tr}\left((\mathbf{W}^V\mathbf{W}^{V\top})^k\right)\right],$$

using simple linear algebra. Under the assumption that the matrices $\mathbf{A}$ and $\mathbf{W}^V$ are asymptotically free, the above limiting moment can be written as the product of individual limiting moments, i.e.

$$\lim_{T,d\to\infty} \mathbb{E}\left[\frac{1}{Td}\operatorname{tr}(\mathbf{J}_{\ell}\mathbf{J}_{\ell}^{\top})^k\right] = \lim_{T\to\infty} \mathbb{E}\left[\frac{1}{T}\operatorname{tr}\left((\mathbf{A}^{\perp}\mathbf{A}^{\perp\top})^k\right)\right] \lim_{d\to\infty} \mathbb{E}\left[\frac{1}{d}\operatorname{tr}\left((\mathbf{W}^V\mathbf{W}^{V\top})^k\right)\right],$$

where each factor equals the $k$-th moment of the limiting squared singular value distribution of its respective matrix. For both $\mathbf{A}^\perp$ and $\mathbf{W}^V$ the limits exist almost surely, and are equal (up to a variance factor) to the well-known Fuss-Catalan numbers, defined by

$$\text{FC}_\ell(k) := \frac{1}{\ell k + 1} \binom{\ell k + k}{k}.$$

Therefore, almost surely,

$$\lim_{T,d\to\infty} \mathbb{E}\Big[\frac{1}{Td}\text{tr}(\mathbf{J}_\ell \mathbf{J}_\ell^\top)^k\Big] = (\sigma_A^2)^k \text{FC}_\ell(k) \times (\sigma_V^2)^k \text{FC}_\ell(k).$$

Simple calculations in the case $k = 1$ and $k = 2$ yield the specified formulae for mean and variance. $\qquad\square$

### A.3. Lemmas

**Lemma A.2.** *Let $\mathbf{W}_1 \in \mathbb{R}^{T\times d}$ and $\mathbf{W}_2, \ldots, \mathbf{W}_q \in \mathbb{R}^{d\times d}$ be independent Gaussian matrices with i.i.d. $\mathcal{N}(0,1)$ entries, and $\mathbf{u} \in \mathbb{R}^T$ a unit vector. Then,*

$$\mathbb{E}\big[s_1^2(\mathbf{u}\mathbf{u}^\top \mathbf{W}_1 \ldots \mathbf{W}_q)\big] = d^q, \tag{22}$$

*and the event*

$$\left| \frac{s_1(\mathbf{u}\mathbf{u}^\top \mathbf{W}_1 \ldots \mathbf{W}_q)}{d^{q/2}} - 1 \right| < t$$

*holds with overwhelming probability.*

*Proof.* First of all, note that the distribution of $s_1(\mathbf{u}\mathbf{u}^\top \mathbf{W}_1 \ldots \mathbf{W}_q)$ is independent of the choice of $\mathbf{u}$, since $\mathbf{W}_1, \ldots, \mathbf{W}_q$ are rotation-invariant. Let us write $\mathbf{u}^\top \mathbf{W}_1 = \alpha_1 \mathbf{u}_1^\top$, where $\mathbf{u}_1 \in \mathbb{R}^d$ has length 1. Similarly, define

$$\alpha_{i+1} := \|\mathbf{u}_i^\top \mathbf{W}_{i+1}\|_2, \quad \mathbf{u}_{i+1}^\top := \frac{\mathbf{u}_i^\top \mathbf{W}_{i+1}}{\alpha_{i+1}},$$

for $1 \le i \le q - 1$. So, we can write

$$\begin{aligned}
s_1(\mathbf{u}\mathbf{u}^\top \mathbf{W}_1 \mathbf{W}_2 \ldots \mathbf{W}_q) &= s_1(\mathbf{u}(\alpha_1 \mathbf{u}_1^\top)\mathbf{W}_2 \ldots \mathbf{W}_q) \\
&= s_1(\mathbf{u}(\alpha_1 \alpha_2 \mathbf{u}_2^\top)) \ldots \mathbf{W}_q) \\
&= \ldots \\
&= \alpha_1 \ldots \alpha_q \cdot s_1(\mathbf{u}\mathbf{u}_q^\top) \\
&= \alpha_1 \ldots \alpha_q,
\end{aligned}$$

where $s_1(\mathbf{u}\mathbf{u}_q^\top) = 1$ since $\mathbf{u}\mathbf{u}_q^\top$ naturally takes the form of an SVD with a single nonzero singular value equal to 1. The random variables $\alpha_1, \ldots, \alpha_q$ are independent (by independence of $\mathbf{W}_i$'s) and identically distributed (by rotation-invariance of $\mathbf{W}_i$'s). Without loss of generality, we can substitute $\mathbf{e}_1$ (the first column of the identity matrix) for $\mathbf{u}$ or $\mathbf{u}_i$ to get

$$\alpha_i \overset{d}{=} \|\mathbf{e}_1^\top \mathbf{W}_i\| = \|\mathbf{w}\|$$

where $\mathbf{w} \in \mathbb{R}^d$ (the first row of $\mathbf{W}_i$) has i.i.d. $\mathcal{N}(0,1)$ entries. Thus, $\mathbb{E}(\alpha_i^2) = \mathbb{E}(\|\mathbf{w}\|_2^2) = d$, and by independence of $\alpha_i$'s we have

$$\mathbb{E}\big[s_1^2(\mathbf{u}\mathbf{u}^\top \mathbf{W}_1 \ldots \mathbf{W}_q)\big] = d^q.$$

Moreover, since each $\alpha_i^2$ has a chi-squared distribution with $d$ degrees of freedom, we can write it as the sum of $d$ independent squared standard Gaussian random variables $\alpha_i^2 = \sum_{j=1}^d w_{i,j}^2$. Thus,

$$s_1^2(\mathbf{u}\mathbf{u}^\top \mathbf{W}_1 \ldots \mathbf{W}_q) = \prod_{i=1}^q \alpha_i^2 = \prod_{i=1}^q (w_{i,1}^2 + \cdots + w_{i,d}^2) = \sum_{j=1}^{d^q} Z_j^2,$$

where each $Z_j$ is the product of $q$ independent $\mathcal{N}(0,1)$ random variables, and therefore is sub-Weibull with parameter $2/q$. We shall apply generalised Bernstein's inequality for the normalised sum of mean-zero sub-Weibull random variables (Kuchibhotla & Chakrabortty, 2022; Bong & Kuchibhotla, 2023), i.e.

$$\mathbb{P}\Big(\Big|\frac{1}{N}\sum_{i=1}^{N} X_i\Big| \geq u\Big) \leq 2\exp\big[-CN\min(\frac{u^2}{K^2}, \frac{u^\beta}{K^\beta})\big], \tag{23}$$

where $X_i$'s are independent mean-zero sub-Weibull random variables with parameter $\beta$ and $K := \max_i \|X_i\|_{\psi_\beta}$. Applying (23) on

$$\frac{1}{d^q}s_1^2(\mathbf{u}\mathbf{u}^\top \mathbf{W}_1\ldots\mathbf{W}_q) - 1 = \frac{1}{d^q}\sum_{j=1}^{d^q}(Z_j^2 - 1),$$

where each $(Z_j^2 - 1)$ is centered sub-Weibull with parameter $1/q$, we get

$$\mathbb{P}\Big(\Big|\frac{1}{d^q}s_1^2(\mathbf{u}\mathbf{u}^\top \mathbf{W}_1\ldots\mathbf{W}_q) - 1\Big| \geq u\Big) = \mathbb{P}\Big(\Big|\frac{1}{d^q}\sum_{j=1}^{d^q}(Z_j^2 - 1)\Big| \geq u\Big)$$
$$\leq 2\exp\big[-C'd^q\min(u^2, u^{1/q})\big],$$

where we have absorbed the dependency on $K = \|(Z_j^2 - 1)\|_{\psi_{1/q}}$ into $C'$. Combining the above with the simple fact that $|z - 1| \geq t$ implies $|z^2 - 1| \geq \max(t, t^2)$, we obtain for any $t \geq 0$ that

$$\mathbb{P}\Big(\Big|\frac{1}{d^{q/2}}s_1(\mathbf{u}\mathbf{u}^\top \mathbf{W}_1\ldots\mathbf{W}_q) - 1\Big| \geq t\Big)$$
$$\leq \mathbb{P}\Big(\Big|\frac{1}{d^q}s_1^2(\mathbf{u}\mathbf{u}^\top \mathbf{W}_1\ldots\mathbf{W}_q) - 1\Big| \geq \max(t, t^2)\Big)$$
$$\leq 2\exp\big[-C'd^q\min(t^2, t^{2/q})\big],$$

i.e. $s_1(\mathbf{u}\mathbf{u}^\top \mathbf{W}_1\ldots\mathbf{W}_q)$ is sub-Weibull with parameter $2/q$ and

$$\Big|\frac{s_1(\mathbf{u}\mathbf{u}^\top \mathbf{W}_1\ldots\mathbf{W}_q)}{d^{q/2}} - 1\Big| < t$$

holds with probability at least $1 - 2\exp\big[-C'd^q\min(t^2, t^{2/q})\big]$, i.e. with *overwhelming* probability (Tao, 2012). $\qquad\square$

**Lemma A.3.** *Consider $p$ i.i.d. Markov matrices $\mathbf{A}_1,\ldots,\mathbf{A}_p \in \mathbb{R}^{T\times T}$ as defined in 3.1, and let $\mathbf{1}_{T\times T}$ be the matrix full of ones. Then, almost surely,*

$$s_1\Big(\mathbf{A}_p\ldots\mathbf{A}_1 - \frac{1}{T}\mathbf{1}_{T\times T}\Big) = O(T^{-p/2}) \tag{24}$$

*Proof.* Let us first show $s_1(\mathbf{A}_p\ldots\mathbf{A}_1) \xrightarrow{a.s.} 1$, as $T$ grows. Each matrix $\mathbf{A}_i$ can be written as the row-normalisation of a table $\mathbf{M}_i$ of i.i.d. random variables, i.e. $\mathbf{A}_i := \mathbf{D}_i\mathbf{M}_i$, where $\mathbf{D}_i$ is a $T \times T$ diagonal matrix containing the inverse row sums of $\mathbf{M}_i$. The entries in $\mathbf{M}_i$ have a finite fourth moment, and, without loss of generality, mean 1 and variance $\sigma^2$. Thus,

$$s_1(T^{p/2}\mathbf{A}_p\ldots\mathbf{A}_1) = s_1(T^{p/2}\mathbf{D}_p\mathbf{M}_p\ldots\mathbf{D}_1\mathbf{M}_1)$$
$$\leq s_1(T\mathbf{D}_p)s_1(T^{-1/2}\mathbf{M}_p)\ldots s_1(T\mathbf{D}_1)s_1(T^{-1/2}\mathbf{M}_1).$$

Following the argument given in (Bordenave et al., 2011), $s_1(T\mathbf{D}_i) = 1 + o(1)$ and $s_1(T^{-1/2}\mathbf{X}_i) \leq \sqrt{T} + O(1)$, for all $1 \leq i \leq p$. Therefore,

$$s_1(T^{p/2}\mathbf{A}_p\ldots\mathbf{A}_1) \leq \big(\sqrt{T} + O(1)\big)^p(1 + o(1))$$
$$\leq T^{p/2}(1 + o(1)),$$

which yields, almost surely, $\lim s_1(\mathbf{A}_p \dots \mathbf{A}_1) \leq 1$. The converse inequality is an immediate consequence of the closure of the set of i.i.d. Markov matrices under matrix multiplication, which gives $\lambda_1(\mathbf{A}_p \dots \mathbf{A}_1) = 1$, combined with $s_1(\mathbf{A}_p \dots \mathbf{A}_1) \geq |\lambda_1(\mathbf{A}_p \dots \mathbf{A}_1)|$. Hence, almost surely, $\lim s_1(\mathbf{A}_p \dots \mathbf{A}_1) = 1$.

Let $\varphi \in \mathbb{R}^T$ be the unit vector such that $\frac{1}{T}\mathbf{1}_{T \times T} = \varphi\varphi^\top$, i.e. $\varphi = T^{-1/2}(1, \dots, 1)^\top$. Also, let $\mathbf{A} := \mathbf{A}_p \dots \mathbf{A}_1$ and define $\mathbf{A}^\perp := \mathbf{A} - \varphi\varphi^\top$. Since the rows of $\mathbf{A}$ sum to 1, our construction ensures that those of $\mathbf{A}^\perp$ sum to zero. We want to show that $s_1(\mathbf{A}^\perp) = s_2(\mathbf{A})\big(1 + o(1)\big)$. To this end, consider the SVD of the matrix $\mathbf{A}^\perp$. There exist orthogonal matrices $\mathbf{U}, \mathbf{V}$ and a diagonal matrix $\mathbf{\Sigma} := \mathrm{diag}\big(s_1(\mathbf{A}^\perp), \dots, s_n(\mathbf{A}^\perp)\big)$ such that

$$\mathbf{A}^\perp = \mathbf{U}\mathbf{\Sigma}\mathbf{V}^\top.$$

Note that since $\mathbf{A}^\perp \varphi = \mathbf{0}$, the matrix has rank at most $T - 1$ and thus $s_n(\mathbf{A}^\perp) = 0$. We will now try to relate the singular values of $\mathbf{A}^\perp$ to those of $\mathbf{A}$, observing that $\mathbf{A}$ is a rank-one perturbation of $\mathbf{A}^\perp$, i.e.

$$\begin{aligned} \mathbf{A} &= \varphi\varphi^\top + \mathbf{A}^\perp \\ &= \varphi\varphi^\top + \mathbf{U}\mathbf{\Sigma}\mathbf{V}^\top. \end{aligned}$$

The squared singular values of $\mathbf{A}$ are exactly the eigenvalues of

$$\mathbf{A}\mathbf{A}^\top = \varphi\varphi^\top + \mathbf{U}\mathbf{\Sigma}^2\mathbf{U}^\top. \tag{25}$$

Since eigenvalues are invariant under orthogonal operators, we can multiply on the left and right by, respectively, $\mathbf{U}^\top$ and $\mathbf{U}$ to get a diagonal matrix perturbed by a rank-one matrix:

$$\mathbf{U}^\top \mathbf{A}\mathbf{A}^\top \mathbf{U} = \mathbf{U}^\top \varphi\varphi^\top \mathbf{U} + \mathbf{\Sigma}^2. \tag{26}$$

Taking the trace, we have

$$s_1^2(\mathbf{A}) + \cdots + s_n^2(\mathbf{A}) = 1 + s_1^2(\mathbf{A}^\perp) + \cdots + s_{n-1}^2(\mathbf{A}^\perp). \tag{27}$$

On the other hand, we can apply Thompson-Lidskii's interlacing inequalities (Thompson, 1976) on Equation (26) to get

$$s_1^2(\mathbf{A}) \geq s_1^2(\mathbf{A}^\perp) \geq s_2^2(\mathbf{A}) \geq s_2^2(\mathbf{A}^\perp) \geq \cdots \geq s_{n-1}^2(\mathbf{A}^\perp) \geq s_n^2(\mathbf{A}) \geq 0. \tag{28}$$

Combining Equations (27) and (28), one obtains

$$s_1^2(\mathbf{A}) + s_2^2(\mathbf{A}) \geq 1 + s_1^2(\mathbf{A}^\perp).$$

As established earlier, almost surely, $\lim s_1(\mathbf{A}) = 1$. So we conclude that in the limit, almost surely, $s_2(\mathbf{A}) \geq s_1(\mathbf{A}^\perp)$. The converse is already given by (28). Therefore we have

$$s_1(\mathbf{A}^\perp) = s_2(\mathbf{A})\big(1 + o(1)\big),$$

almost surely. Note that the same reasoning is valid for the case $p = 1$, and results in $s_1(\mathbf{A}_i^\perp) \xrightarrow{a.s.} s_2(\mathbf{A}_i)$ for any $i$.

Having shown the convergence of the largest singular value of $\mathbf{A}^\perp$ to the second largest singular value of $\mathbf{A}$, we now show that $s_2(\mathbf{A})$ is of order $T^{-p/2}$. To this end, note that the matrix can be written as a rank-one perturbation of the product of $\mathbf{A}_i^\perp$'s, i.e.

$$\begin{aligned} \mathbf{A} &= \mathbf{A}_p \dots \mathbf{A}_1 \\ &= (T^{-1}\mathbf{1}_{T \times T} + \mathbf{A}_p^\perp) \dots (T^{-1}\mathbf{1}_{T \times T} + \mathbf{A}_1^\perp) \\ &= T^{-1}\mathbf{1}_{T \times T}\big(\mathbf{I} + \mathbf{A}_1^\perp + \cdots + \mathbf{A}_{p-1}^\perp \dots \mathbf{A}_1^\perp\big) + \mathbf{A}_p^\perp \dots \mathbf{A}_1^\perp, \end{aligned}$$

where some of the terms vanish since $\mathbf{A}_i^\perp \varphi = \mathbf{0}$. Given that $\mathrm{rank}(\mathbf{A} - \mathbf{A}_p^\perp \dots \mathbf{A}_1^\perp) = 1$, we can apply Thompson-Lidskii's inequality to get

$$s_1(\mathbf{A}_p^\perp \dots \mathbf{A}_1^\perp) \geq s_2(\mathbf{A}).$$

By submutiplicativity of the operator norm, this implies $s_1(\mathbf{A}_p^\perp)\ldots s_1(\mathbf{A}_1^\perp) \geq s_2(\mathbf{A})$. Moreover, we previously established that for each individual matrix $\mathbf{A}_i$, $s_1(\mathbf{A}_i^\perp) \xrightarrow{a.s.} s_2(\mathbf{A}_i)$, and it is shown in (Bordenave et al., 2011) that $s_2(\mathbf{A}_i) \xrightarrow{a.s.} 2\sigma T^{-1/2}$. Therefore, we conclude that

$$s_2(\mathbf{A}) \leq \left(2\sigma T^{-1/2}\right)^p = O(T^{-p/2}).$$

Combined with Equation (A.3), we have

$$s_1(\mathbf{A} - \frac{1}{T}\mathbf{1}_{T\times T}) = s_1(\mathbf{A}^\perp) = O(T^{-p/2}),$$

almost surely. □

**Theorem A.4.** *Let $\mathbf{A}_1,\ldots,\mathbf{A}_p \in \mathbb{R}^{T\times T}$ be independent i.i.d. Markov matrices as defined in 3.1 and $\mathbf{W}_1 \in \mathbb{R}^{T\times d}$, $\mathbf{W}_2,\ldots,\mathbf{W}_q \in \mathbb{R}^{d\times d}$ be independent Gaussian matrices with i.i.d. $\mathcal{N}(0,1)$ entries. Then, for large enough $T$ and $d$ with fixed $\gamma = T/d \in (0,1]$, the event*

$$\left|\frac{s_1(\mathbf{A}_p\ldots\mathbf{A}_1\mathbf{W}_1\ldots\mathbf{W}_q)}{d^{q/2}} - 1\right| < t,$$

*holds with overwhelming probability.*

*Proof.* We write $\mathbf{A} := \mathbf{A}_p\ldots\mathbf{A}_1 = \varphi\varphi^\top + \mathbf{A}^\perp$ and $\mathbf{W} := \mathbf{W}_1\ldots\mathbf{W}_q$. Then, using the triangle inequality $|s_1(A) - s_1(B)| \leq s_1(A+B) \leq s_1(A) + s_1(B)$, we have

$$|s_1(\varphi\varphi^\top\mathbf{W}) - s_1(\mathbf{A}^\perp\mathbf{W})| \leq s_1(\mathbf{A}\mathbf{W}) = s_1(\varphi\varphi^\top\mathbf{W} + \mathbf{A}^\perp\mathbf{W})$$
$$\leq s_1(\varphi\varphi^\top\mathbf{W}) + s_1(\mathbf{A}^\perp\mathbf{W}).$$

On the other hand, it is well known that the largest singular value of a Gaussian matrix converges almost surely to the soft edge of the bulk of the limiting density (Geman, 1980), i.e.

$$s_1\left(\frac{1}{\sqrt{d}}\mathbf{W}_i\right) \xrightarrow{a.s.} \begin{cases} 1 + \sqrt{\gamma}, & i = 1, \\ 2, & i \geq 2. \end{cases}$$

Therefore, by submultiplicativity of $s_1$, we have

$$s_1(\mathbf{W}) \leq s_1(\mathbf{W}_1)\ldots s_1(\mathbf{W}_q) \leq \left(2\sqrt{d} + o(\sqrt{d})\right)^q = 2^q d^{q/2} + o(d^{q/2}). \tag{29}$$

Combining (29) with Lemma A.3, we get

$$s_1(\mathbf{A}^\perp\mathbf{W}) \leq s_1(\mathbf{A}^\perp)s_1(\mathbf{W}) = O(d^{\frac{q-p}{2}}), \tag{30}$$

and thus, almost surely,

$$\left|s_1(\varphi\varphi^\top\mathbf{W}) - O(d^{\frac{q-p}{2}})\right| \leq s_1(\mathbf{A}\mathbf{W}) \leq s_1(\varphi\varphi^\top\mathbf{W}) + O(d^{\frac{q-p}{2}}).$$

Now, using Lemma A.2, we can assert that $s_1(\varphi\varphi^\top\mathbf{W})$ is close to $d^{q/2}$ with overwhelming probability, i.e.

$$\frac{s_1(\varphi\varphi^\top\mathbf{W})}{d^{q/2}} \in (1-t, 1+t),$$

with a probability greater than $P_{t,d} := 1 - 2\exp\left[-C'd^q \min(t^2, t^{2/q})\right]$. Moreover, by (30),

$$\frac{s_1(\mathbf{A}^\perp\mathbf{W})}{d^{q/2}} \to 0,$$

as $d$ grows. Thus, we can make the above quantity smaller than any given $\varepsilon$. Altogether, for large enough $T$ and $d$, the probability that

$$\left|\frac{s_1(\mathbf{A}\mathbf{W})}{d^{q/2}} - 1\right| < t + \varepsilon$$

is at least $P_{t,d}$. Since $\varepsilon$ is arbitrary the proof is complete.

□

**Theorem A.5.** *Let $\mathbf{A}_1, \ldots, \mathbf{A}_p \in \mathbb{R}^{T \times T}$ be i.i.d. Markov matrices as defined in 3.1 and $\mathbf{W}_1 \in \mathbb{R}^{T \times d}$, $\mathbf{W}_2 \ldots, \mathbf{W}_q \in \mathbb{R}^{d \times d}$ be independent Gaussian matrices with i.i.d. $\mathcal{N}(0, 1)$ entries. Then, for $T$ and $d$ large enough,*

$$s_2(\mathbf{A}_p \ldots \mathbf{A}_1 \mathbf{W}_1 \ldots \mathbf{W}_q) = O(d^{\frac{q-p}{2}}). \tag{31}$$

*Proof.* To exhibit a spectral gap in $\mathbf{AW}$, it suffices to bound its second largest singular value by a quantity significantly lower than where the largest singular value is concentrated. To this end, observe that $\mathbf{AW}$ is a rank-one perturbation of $\mathbf{A}^{\perp}\mathbf{W}$:

$$\mathbf{AW} = (\mathbf{A}^{\perp} + \varphi\varphi^{\top})\mathbf{W} = \mathbf{A}^{\perp}\mathbf{W} + \varphi\varphi^{\top}\mathbf{W}.$$

Thus, using Weyl's inequality, we can write

$$s_2(\mathbf{AW}) \leq s_1(\mathbf{A}^{\perp}\mathbf{W}) + s_2(\varphi\varphi^{\top}\mathbf{W}) = s_1(\mathbf{A}^{\perp}\mathbf{W}).$$

Next, by submultiplicativity of the operator norm combined with upper bounds in Lemma A.3 and(29),

$$s_1(\mathbf{A}^{\perp}\mathbf{W}) \leq s_1(\mathbf{A}^{\perp})s_1(\mathbf{W}) = O(T^{-p/2})O(d^{q/2}).$$

Therefore,

$$s_2(\mathbf{AW}) = O(d^{\frac{q-p}{2}}).$$

$\square$

**Lemma A.6** (Bulk distribution of $\mathbf{A}^{\perp}$). *Let $\mathbf{A} \in \mathbb{R}^{T \times T}$ be an i.i.d. Markov matrix, and let $\mathbf{A}^{\perp} := \mathbf{A} - T^{-1}\mathbf{1}_{T \times T}$. Then, almost surely, the empirical singular value distribution of $T^{1/2}\mathbf{A}^{\perp}$ weakly converges to the quartercircular law as $T \to \infty$, i.e.*

$$\nu_{\sqrt{T}\mathbf{A}^{\perp}} := \frac{1}{T}\sum_{i=1}^{T}\delta_{s_i(\sqrt{T}\mathbf{A}^{\perp})} \xrightarrow{\mathcal{C}_b} \mathcal{Q}_{\sigma}, \tag{32}$$

*where $\mathcal{Q}_{\sigma}$ is the quartercircular law on the real interval $[0, 2\sigma]$ with Lebesgue density*

$$x \mapsto \frac{1}{\pi\sigma^2}\sqrt{4\sigma^2 - x^2}\mathbb{1}_{[0,2\sigma]}.$$

*Moreover, almost surely, $\mathbf{A}^{\perp}$ does not exhibit any spectral gap.*

*Proof.* Thompson-Lidskii's interlacing result for finite rank perturbation (Thompson, 1976) states that for any $n \times n$ matrices $\mathbf{M}$ and $\mathbf{M}'$ with $\text{rank}(\mathbf{M} - \mathbf{M}') \leq k$, we have

$$s_{i-k}(\mathbf{M}) \leq s_i(\mathbf{M}') \leq s_{i+k}(\mathbf{M}).$$

This in turn yields the following bulk inequality,

$$\|F_{\mathbf{M}} - F_{\mathbf{M}'}\|_{\infty} \leq \frac{\text{rank}(\mathbf{M} - \mathbf{M}')}{n},$$

where $F_{\mathbf{M}}$ and $F_{\mathbf{M}'}$ denote the cumulative distribution functions of $\nu_{\mathbf{M}}$ and $\nu_{\mathbf{M}'}$, respectively. Since $\text{rank}(\mathbf{A} - \mathbf{A}^{\perp}) = 1$, then

$$\|F_{\sqrt{T}\mathbf{A}} - F_{\sqrt{T}\mathbf{A}^{\perp}}\|_{\infty} \leq \frac{1}{T} \xrightarrow[T \to \infty]{} 0.$$

Combining the above limit with the fact that $\nu_{\sqrt{T}\mathbf{A}} \xrightarrow{\mathcal{C}_b} \mathcal{Q}_{\sigma}$ almost surely (see (Bordenave et al., 2011)), we deduce that

$$\nu_{\sqrt{T}\mathbf{A}^{\perp}} \xrightarrow{\mathcal{C}_b} \mathcal{Q}_{\sigma}$$

almost surely. The almost sure absence of outliers in the singular value distribution of $\mathbf{A}^{\perp}$ can be immediately inferred from Lemma A.3 when $p = 1$. $\square$

**Lemma A.7.** *Let $0 < \sigma_i < \infty$ and $0 < \gamma_i \leq 1$ for $1 \leq i \leq n$. Let $\mathcal{M}$ be the free multiplicative convolution of $\mathcal{MP}(\gamma_i, \sigma_i)$ distributions, i.e.*

$$\mathcal{M} := \mathcal{MP}(\gamma_1, \sigma_1) \boxtimes \mathcal{MP}(\gamma_2, \sigma_2) \boxtimes \cdots \boxtimes \mathcal{MP}(\gamma_n, \sigma_n).$$

*Then the mean and variance of $Z \sim \mathcal{M}$ are given by*

$$\mathbb{E}(Z) = \prod_{i=1}^{n} \sigma_i^2, \tag{33}$$

$$\text{Var}(Z) = \Big( \prod_{i=1}^{n} \sigma_i^2 \Big)^2 \big( \gamma_1 + \gamma_1 \gamma_2 + \cdots + \gamma_1 \gamma_2 \cdots \gamma_n \big). \tag{34}$$

*Proof.* The distribution in question $\mathcal{M}$ is the limiting squared singular value distribution of a product of rectangular independent Gaussian matrices, whose general moments are worked out in (Akemann et al., 2013). Simple algebraic manipulations lead to our result. □

## A.4. Can transformers achieve dynamical isometry?

In Section 3, we have established (i) the existence of an outlier eigenvalue/singular value in the spectrum of softmax-based attention matrices, and (ii) that removing this outlier helps with rank collapse and exploding gradients. In the absence of the outlier, we can take a further step to analyse the bulk of the spectra of the network's token-wise covariance and input-output Jacobian—two quantities that have been shown to play a key role in signal propagation as we will see below. To this end, we will make use of free probability theory.

**Free probability.** The theory of free probability studies "non-commuting random variables" such as random matrices (see (Mingo & Speicher, 2017) for a textbook introduction). Pioneered by (Pennington et al., 2017; 2018), the theory has found powerful applications in the analysis of large random neural networks. Notably, it provides tools to characterise the singular value distribution of sums or products of random matrices. Loosely speaking, "freeness" plays the same role for random matrices as independence does for (scalar) random variables. Freeness allows us to compute the limiting spectral density of a product $\mathbf{M}_n \mathbf{M}'_n$ from the limiting spectral densities of $\mathbf{M}_n$ and $\mathbf{M}'_n$, just as independence enables the computation, for instance, of the moments of $ZZ'$, given those of $Z$ and $Z'$. Specifically, if $\nu_{\mathbf{M}_n} \to \nu$, $\nu_{\mathbf{M}'_n} \to \nu'$, and $\mathbf{M}_n$ and $\mathbf{M}'_n$ are *asymptotically* free, then

$$\nu_{\mathbf{M}_n \mathbf{M}'_n} \xrightarrow{n \to \infty} \nu \boxtimes \nu',$$

where $\boxtimes$ denotes an operation called *free multiplicative convolution*.

Let us assume that the input tokens are orthonormal, i.e. $\mathbf{\Sigma}_0 = \mathbf{X}_0 \mathbf{X}_0^\top = \mathbf{I}$. As a criterion for faithful signal propagation, one should require that $\mathbf{\Sigma}_\ell^\perp$ stay close to the identity matrix. Considering the spectrum, this means that the limiting singular value distribution of $\mathbf{\Sigma}_\ell^\perp$ should concentrate around the value 1. A natural approach, as demonstrated in the fully-connected case in (Pennington et al., 2017; 2018; Murray et al., 2022), is to adjust the model's hyperparameters to ensure that the mean of the limiting distribution is $O(1)$ and the variance is minimised. Proposition A.8 describes the moments of the limiting singular value distribution of $\mathbf{\Sigma}_\ell^\perp$.

**Proposition A.8** (Bulk of covariance kernel's singular value distribution). *Let $\mathbf{X}_\ell^\perp = \mathbf{A}_\ell^\perp \mathbf{X}_{\ell-1}^\perp \mathbf{W}_\ell^V$ be the signal at layer $\ell$ in our modified model* (16) *and $\mathbf{\Sigma}_\ell^\perp = \mathbf{X}_\ell^\perp \mathbf{X}_\ell^{\perp \top} \in \mathbb{R}^{T \times T}$ be its covariance matrix. Let the underlying i.i.d. Markov matrices $\mathbf{A}_\ell$ have variance $\sigma_A^2$ and $\mathbf{W}_\ell^V$ have i.i.d. $\mathcal{N}(0, \sigma_V^2)$ entries. Let $\mathbf{\Sigma}_0^\perp = \mathbf{I}$ and $\mathcal{D}_\ell$ be the limiting singular value distribution of $\mathbf{\Sigma}_\ell^\perp$. Then the mean and variance of $Z \sim \mathcal{D}_\ell$ are given by*

$$\mathbb{E}(Z) = (\sigma_A \sigma_V / \sqrt{\gamma})^{2\ell}, \tag{35}$$

$$\text{Var}(Z) = \ell(1 + \gamma)(\sigma_A \sigma_V / \sqrt{\gamma})^{4\ell}, \tag{36}$$

*where $\gamma := \frac{T}{d} \in (0, 1]$.*

The assumption $\gamma \leq 1$ is not essential and is made only to ensure that $\mathbf{\Sigma}_\ell^\perp$ is full-rank, avoiding trivial zero singular values. If $\gamma > 1$, then the limiting singular value distribution is given by $(1 - \gamma^{-1})\delta_0 + \gamma^{-1}\mathcal{D}_\ell$ and the mean and variance should be adjusted accordingly.

It is evident from the above proposition that simultaneously controlling both the mean and variance of $\mathcal{D}_\ell$ is not feasible. Model (16) does not have enough hyperparameters to achieve this balance. Indeed, to prevent the mean from growing or shrinking exponentially with depth, the product $\sigma_A \sigma_V$ must equal $\sqrt{\gamma}$. However, this constraint leads to the variance increasing linearly with $\ell$.

The Jacobian of the input-to-output function $f : \mathbf{X}_0 \mapsto \mathbf{X}_L^\perp$, represented by our modified transformer model, characterises the network's sensitivity to input perturbations up to first order, according to

$$f(\mathbf{X}_0 + \epsilon \mathbf{U}) \approx f(\mathbf{X}_0) + \epsilon \frac{\partial f}{\partial \mathbf{X}}\bigg|_{\mathbf{X}_0} \mathbf{U}. \tag{37}$$

Let us consider the matricised version of the Jacobian at layer $\ell$, i.e.

$$\mathbf{J}_\ell := \frac{\partial \operatorname{vec}(\mathbf{X}_\ell^\perp)}{\partial \operatorname{vec}(\mathbf{X}_0)} \in \mathbb{R}^{Td \times Td}. \tag{38}$$

The goal is to ensure that the spectral energy of the Jacobian concentrates around 1, thereby minimising distortion of the input space geometry—a property often referred to as the *dynamical isometry* in the literature (see (Pennington et al., 2017)). For our model (16), it is straightforward to show

$$\mathbf{J}_\ell = (\mathbf{A}_\ell^\perp \cdots \mathbf{A}_1^\perp) \otimes (\mathbf{W}_1^V \cdots \mathbf{W}_\ell^V) \in \mathbb{R}^{Td \times Td}, \tag{39}$$

where $\otimes$ denotes the Kronecker product. Proposition A.9 describes the moments of the limiting squared singular value distribution of $\mathbf{J}_\ell$.

**Proposition A.9** (Bulk of Jacobian's squared singular value distribution). *Let* $\mathbf{X}_\ell^\perp = \mathbf{A}_\ell^\perp \mathbf{X}_{\ell-1}^\perp \mathbf{W}_\ell^V$ *be the signal at layer* $\ell$ *in our modified model* (16). *Let the underlying i.i.d. Markov matrices* $\mathbf{A}_\ell$ *have variance* $\sigma_A^2$ *and* $\mathbf{W}_\ell^V$ *have i.i.d.* $\mathcal{N}(0, \sigma_V^2)$ *entries. Let* $\mathcal{D}_\ell$ *be the limiting distribution of the squared singular values of* $\mathbf{J}_\ell := \partial \mathbf{X}_\ell^\perp / \partial \mathbf{X}_0$[4]. *Then the mean and variance of* $Z \sim \mathcal{D}_\ell$ *are given by*

$$\mathbb{E}(Z) = (\sigma_A \sigma_V)^{2\ell}, \tag{40}$$

$$\operatorname{Var}(Z) = \ell(\ell + 2)(\sigma_A \sigma_V)^{4\ell}. \tag{41}$$

Controlling the mean leads to a quadratically growing variance, while minimising the variance is only achievable if $\sigma_A \sigma_V < 1$, which, in turn, causes the mean to vanish. Without considering a more complex model, no choice of $(\sigma_A, \sigma_V)$ can achieve our goal of dynamical isometry.

### A.5. Implementation details

**Architecture.** The default model consists of a stack of single-head attention layers, with an optional LayerNorm inserted between them (denoted by "+ LN" in the legend) after receiving an optional skip connection from the previous layer (denoted by "+ skip" in the legend). When both options are enabled simultaneously, the configuration is referred to as "+ skip + LN". By single-head, we mean that only one attention mechanism is computed, applied to the values and then multiplied by a matrix $\mathbf{W}_h$ which is initialised as the identity matrix.

**Attention design.** At initialisation, when the attention is labelled as "$\mathbf{A}$", the matrix is sampled from the set of i.i.d. Markov matrices, as defined in Definition 3.1, with a variance of $\sigma_A = 1$. To achieve this, we sample a random matrix $\mathbf{B}$ with i.i.d. lognormal entries and apply softmax row-wise such that $\mathbf{A} := \operatorname{softmax}(\mathbf{B})$. The moments of $\mathbf{B}$ are adjusted precisely so that $\sigma_A = 1$. If "Identity $\mathbf{A}$" is chosen, the attention matrix is a constant equal to the identity. When the attention is labelled as "$\mathbf{A}(\mathbf{X})$", the key/query matrices are sampled from i.i.d. Gaussian matrices $\mathcal{N}(0, 1)$ and the standard key-query softmax-attention matrix is formed. If a label indicates a "$\perp$", the forward pass of the attention mechanism is systematically (at initialisation) adjusted so that the spectral gap is removed, as in our modified model (16). If the label indicates "no outliers", then, in the forward pass, an SVD is performed, and the $r$ largest singular values are taken out as follows:

$$\mathbf{A}^{\text{no outliers}} = \mathbf{A} - \sum_{i=1}^r s_i u_i v_i^\top,$$

---

[4]In a minor abuse of notation, we may write $\partial \mathbf{X}_\ell^\perp / \partial \mathbf{X}_0$ as a shorthand for $\partial \operatorname{vec}(\mathbf{X}_\ell^\perp) / \partial \operatorname{vec}(\mathbf{X}_0)$.

where $\mathbf{A} = \mathbf{U}\mathrm{diag}\big(s_1(\mathbf{A}), \ldots, s_n(\mathbf{A})\big)\mathbf{V}^\top$. The cut off threshold $r$ is chosen such that it maximises the difference between consecutive singular values, i.e.

$$r \coloneqq \underset{1 \leq i \leq n-1}{\arg\max}\,(s_i - s_{i+1}).$$

