# OpenReview forum: "Mind the Gap: a Spectral Analysis of Rank Collapse and Signal Propagation in Attention Layers"
_ICML.cc/2025/Conference — ICML 2025 poster_

### Official Review · Reviewer_j3Cu · 2025-02-27

**Overall Recommendation:** 3

**Summary:**

In this paper, the authors discuss the phenomenon of rank collapse in width (i.e., for asymptotically large context length) in transformers at initialization. The authors point at the spectral gap in the attention matrix as the main cause of such collapse, and they devise a simple solution to remove the gap and tame the rank collapse.

**Claims And Evidence:**

The theorems are stated clearly, and experiments are carried out to validate them. However, I feel that the authors are quite a bit overselling their findings, or at least they are not clearly stating the limitations of their analysis. The main limitations that I find a bit difficult to justify are:
1. The input matrix should be orthonormal, which is quite unrealistic in practical scenarios, since usually the vocabulary size is some 10x the embedding dimension.
2. That the ratio between context length and embedding dimension ($gamma$ in the paper) is a constant smaller than 1, again quite unrealistic in practice (LLama3, e.g., allows context lengths that are more than 10x the embedding dimension).
3. In the analysis for more than one layer, the authors remove the dependence of attention matrices from the input.
I understand that in theory papers such assumptions are usually necessary. However, I also think that such limitations should be clearly stated and discussed in the paper, proposing possible solutions if possible, or stating why it would be hard to overcome them.

**Essential References Not Discussed:**

Other works on rank collapse in depth could be cited, such as:
- Feng et al., "Rank diminishing in deep neural networks," 2022.
- Geshkovski et al., "A mathematical perspective on transformers", 2023.
- Geshkovski et al., "The emergence of clusters in self-attention dynamics", 2023.
- Geshkovski et al., "Dynamic metastability in the self-attention model", 2024.

**Experimental Designs Or Analyses:**

I think that the second main limitation of the paper is the experimental section, both regarding the experiments presented in the paper, and additional experiments that I think are important but missing:
1. Regarding the experiments already present, details are missing about the setups used. For example, it is not clear what are the dimensions of the matrices involved in Fig. 3, nor what is the value of $\gamma$. Without such information, it is hard to evaluate the prominence of the rank collapse, also because the stable rank seems to be already very small from the start (approx. 1.25) compared to what I expect the size of the involved matrices to be.
2. I believe that the following very important experiments are missing, without which it is hard to understand the impact of the work:
    a) Does your proposed solution to the rank collapse problem also work in real transformers models? That is, how would the curves of Figure 3 behave if your solution of removing the one-rank perturbation is applied?
    b) Does the rank collapse in width also occur when many consecutive layers are used? That is, can you show an equivalent of Figure 5 with the stable rank as the y-axis?
    c) Does the rank collapse and/or the gradient explosion also occur when the attention matrices in consecutive layers are kept as layer dependent? That is, in the case of many successive layers, if the usual formula (1) is used for the attention matrices instead of considering them as independent of the input, does the rank collapse in width still occur?
I think these proposed experiments should be easy and quick to implement in your experimental setup, and they would greatly enhance the understanding of the scope of your observed phenomenon and of you proposed solution to it.

**Methods And Evaluation Criteria:**

The proof techniques and experimental setting are sensible.

**Other Comments Or Suggestions:**

N/A

**Other Strengths And Weaknesses:**

I believe that the work is a potentially interesting new direction for the field, if the authors are able to better justify their analysis. In particular, I would like the authors to discuss the theoretical and experimental limitations that I outlined in the sections "Claims and Evidence" and "Experimental Designs Or Analyses" above.

**Questions For Authors:**

Please see sections "Claims And Evidence" and "Experimental Designs Or Analyses" above. I will be happy to revise my score if my concerns are addressed satisfactorily. Again, my main concern is that it is not clear from the provided experiments if the rank collapse in width is a serious issue in practical scenarios, and if the proposed solution helps in that regard.

**Relation To Broader Scientific Literature:**

The paper falls into the line of work investigating rank collapse in transformers. Previous literature focused on rank collapse in depth, i.e., as the number of consecutive attention layers go to infinity. This paper, instead, investigates rank collapse in width.

**Theoretical Claims:**

I checked the proofs at high level and I could not find any major issue that could invalidate the theoretical results presented in the paper.

---

> ### Author Rebuttal · Authors · 2025-04-01
>
> We thank the reviewer for their thorough comments and their appreciation of our contribution. We address the points in the same order. For additional figures (indicated by Roman numerals), please see https://anonymous.4open.science/r/spectral_analysis_transformers-C633/figures_rebuttal.pdf.
>
> 1. Thank you for bringing this up. We have addressed this in detail in our response to reviewer uxct, providing extensive illustrations showing that our findings hold in practice, even without these assumptions, which are only needed for our proofs. We will make sure to clarify these theoretical limitations in our revised version.
>
> 2. Please see the response regarding $\gamma$ in the answer to reviewer uxct. A precise quantitive characterisation of the spectral bulk without these assumptions is extremely interesting and we will explore it, but this is a multi-year programme of study and we believe it is best that the results we have so far be presented to the machine learning community sooner than later. Regarding Llama3, it is interesting to consider the case of a decoder-like architecture, for which an extra layer of complexity comes in due to the causal mask in the attention, which thus falls outside the scope of this paper. Thus, we kindly refer you to our paragraph GPT-architecture in our answer to uxct.
>
> 3. Thank you for highlighting that this was not clearly explained.  The reason we do not include input dependence in later layers is that the resulting inputs do not satisfy the orthogonality conditions and, as mentioned above, a rigorous analysis in the non-orthogonality setting is beyond the scope of what can be theoretically justified at this stage. We will make sure to include a clear discussion on this limitation of our work.
>
> ---
>
> 1. Apologies for these parameters having not been clearly stated. The parameters are now explicitly stated in the revised manuscript, $d=768$ with input length as sentences from our own abstract that are stacked together before being processed with a pre-trained tokeniser that comes with Hugging Face's checkpoints of the corresponding model. All experiments are repeated $5$ times, with the average result shown as a solid line and individual runs displayed as faded lines.
>
> 2. We sincerely thank the reviewer for these interesting suggestions which we believe enhanced our manuscript. We have undertaken these experiments and are happy to share the results with you. (a) We show in Fig. V how rank collapse in width is affected when modifying the information propagation in RoBERTa according to our proposed fix. RoBERTa is such that $d=768$. Whilst this small improvement might seem marginal in width (and the discrepancy with our theoretical guarantees can be explained by the extensive complexity of the architecture beyond our theoretical framework), its impact on the rank collapse in depth is remarkable, see Fig. VI. (b) The equivalent of Fig. 5 in our draft with the y-axis being the stable rank for consecutive layers following equation (1) is exactly Fig. 4b in our submission. The label '$\mathbf{A}(\mathbf{X})$' indicates that information flows as in equation (1), whereas '$\mathbf{A}$' (e.g. in Fig. 5) is used when the attention matrix is taken as an i.i.d. Markov matrix, independent of the input. We realise this distinction was only clearly stated in Appendix A.4 and have moved this up to the main body in the revised version. Nevertheless, rank collapse occurs across consecutive layers, as shown in Fig. 4b, and the only modules that address it—among LayerNorm and skip connections—do so by removing the spectral gap. In case the reviewer was interested in seeing the same plot with i.i.d. Markov matrices instead, we upload an additional figure for which similar comments can be made; see Fig. VII. (c) Fig. 4b in our current submission seems to be what the reviewer is asking for. Regarding the gradients in deeper key-query attention layers, we provide an additional plot in Fig. VIII that reaffirms our analysis and give us some empirical insights on the relationship between rank collapse in width, in depth, and gradients explosion.
>
> ---
>
> + Thank you for highlighting these manuscripts, which we had not originally cited. We will include citations to them in our revised manuscript.
>
> We have taken into consideration your overall impression that the limitations of our work were not clearly stated enough and have modified our draft accordingly. Should you have any further concerns that might affect your appreciation of our work, we would be happy to address them.

---

> > ### Comment · Reviewer_j3Cu · 2025-04-04
> >
> > I thank the authors for the careful response and the additional experiments provided. I think I miswrote some of my requests for experiments. In particular, I would like to ask if the authors would be able to provide experimental results for rank collapse in width after several layers. For example, a plot where the x-axis is $T$ and the y-axis is the stable rank of $\Sigma_{\ell}$ for various values of $\ell$. Please tell me if my request makes sense or if I have misunderstood something.

---

> > > ### Author Response · Authors · 2025-04-06
> > >
> > > We thank the reviewer for acknowledging our responses, which included both clarifications and additional experiments—and, most importantly, for engaging with us. We now provide the requested additional experiment, which can be found as Figures X and XI from https://anonymous.4open.science/r/spectral_analysis_transformers-C633/figures_rebuttal_v2.pdf. As before, figures from this document will be referred to with roman numerals while figures from the original submission are referred to using standard numerals.
> > >
> > > In this experiment, the input has a stable rank exactly equal to T. After a single application of the softmax layer (ℓ=1), the stable rank drops significantly, quickly approaching 1 as T increases. Removing the spectral outlier successfully mitigates rank collapse in width (Fig XI), consistent with our theory. As pointed out by the reviewer, interesting insights can be drawn from deeper layers. After just one application of softmax, the stable rank has already collapsed and is already so low that it leaves little room for further degradation with increasing width when ℓ>1 (Fig. X). Moreover, while deeper layers introduce more complex behavior that our theory is not able to predict, we observe that the proposed fix, which is provably effective for ℓ=1 retains some degree of effectiveness when ℓ>1 (Fig. XI). We thank the reviewer for suggesting this insightful additional experiment and remain happy to address any further.
> > >
> > > We also take this opportunity to summarize our rebuttal:
> > >
> > > - Regarding reviewer’s concerns about the validity of the assumptions that are used in our proofs, we clarified that our work should be seen as a proof of concept, with broader applicability beyond the current framework. Specifically, we:
> > >
> > >   - Conducted an extensive ablation study to show that our findings on the spectral gap within the attention matrix (and subsequent rank collapse in width) hold even without orthogonality constraints on the inputs, both for synthetic (Fig IX) and real world data (Fig I)
> > >   - Provided an ablation study on $\gamma$, defined as the ratio between the number of tokens $T$ and the embedding dimension $d$, to show this assumption is artificial (without it, the inputs can not be assumed to be isometric) and is not needed in practice both on synthetic data (Fig. IX) and real world data (Fig. III)
> > >
> > > - In response to concerns about the generalisability of our findings beyond attention layers, we:
> > >
> > >   - Demonstrated that the spectral gap induced by the softmax attention layer cannot be leveraged by additional modules like skip connections or LayerNorm (Fig. 4) nor by any other component typically found in Bert encoders, as we expicitly show the occurrence of rank collapse within real world transformers (Fig. 3).
> > >
> > >   - Showed that the simple fix we propose mitigates rank collapse in real world transformers using real world data in width (Fig. V), significantly solving rank collapse in depth (Fig. VI).
> > >
> > > - In response to reviewers questions on:
> > >
> > >    - GPT architectures: We showed that a similar spectral gap emerges in GPT/decoder-like architectures (Fig. IV), but occurs on the real line rather than the complex plane due to the causal mask. While our theory does not currently cover this case, it could be extended to accommodate it in a follow-up work, highlighting the utility of our proof of concept.
> > >
> > >    - The connection to existing literature: We clarified why the proposed fixes for rank collapse in depth in [1] and [2] are not suitable for addressing rank collapse in width and we and connected our theoretical insights to the recent practical works [3] and [4].
> > >
> > > We tried our best to answer all the reviewer’s questions and concerns but would be more than happy to answer any further points. We hope the reviewers will reconsider their scores in light of these responses, and we thank them once again for their valuable feedback.
> > >
> > > ---
> > >
> > > [1] Dong, Y. et al. (2021). Attention is not all you need: Pure attention loses rank doubly exponentially with depth.
> > >
> > > [2] Noci, L. et al. (2022). Signal propagation in transformers: Theoretical perspectives and the role of rank collapse.
> > >
> > > [3] Ye, T. et al. (2024). Differential transformer.
> > >
> > > [4] Ali, A. et al. (2023). Centered self-attention layers.

---

### Official Review · Reviewer_W8Pc · 2025-03-11

**Overall Recommendation:** 3

**Summary:**

The authors study attention layers randomly initialized, looking at signal propagation or exploding/vanishing gradient issues from a rank perspective. Notably, using random matrix theory tools, they identify a new rank collapse that occurs in width, i.e., in the context size. Via a careful theoretical analysis, they showcase how it is related to the exploding gradient problem and propose a practical fix to solve the rank collapse in width. They conduct experiments with synthetic inputs on toy and real attention layers, e.g., from BERT, to validate their theoretical findings.

**Claims And Evidence:**

The theoretical claims are supported by clear and detailed proofs, and the authors also provide experimental validation of their theory with synthetic inputs. However, the assumptions (orthogonal inputs) seem unrealistic, which impact the theoretical findings, in my opinion.

**Essential References Not Discussed:**

To the best of my knowledge, there were no essential references not discussed in the current submission.

A potentially relevant paper is [1], which studies the limitations of transformers in time series forecasting by looking at the loss landscape, rank, and entropy collapse. The authors proposed SAMformer, which solves the loss's sharpness that leads to a significant improvement with SOTA methods, and in contrast, they showed that entropy collapse was benign. In the paper, they showcase block diagonal attention matrices with SAMformer (i.e., high rank), while the vanilla Transformer and other modifications suffer from rank collapse (Fig 6 and 12). The connection with the current submission is that the models considered are single-layer transformers; hence, the rank collapse does not come from the depth. In addition, the channel-wise attention means that the number of tokens is the number of features (multivariate time series input); hence, the context is high (up to  862), which could qualify as a rank collapse in "width". [1] could, for instance, be used as motivation for the current submission to study the rank in width (see weakness part) or offer other perspectives for the current study.

*References*

[1] Ilbert et al., SAMformer: Unlocking the Potential of Transformers in Time Series Forecasting with Sharpness-Aware Minimization and Channel-Wise Attention

**Experimental Designs Or Analyses:**

I checked the soundess of the experiments. I believe that, while most of the contributions are theoretical, the experiments are too lightweight with a focus on small-scale, synthetic data and attention-only models. I think the submission would benefit from experiments in more practical settings and/or better empirical motivation to study the rank in width.

**Methods And Evaluation Criteria:**

The authors provide theoretical results to better understand rank collapse in width. They propose a practical fix and conduct synthetic experiments to validate their findings. Hence, the method and evaluation criteria make sense for the problem at hand, although the experiments should illustrate the benefits of the approach to improve signal propagation and/or mitigate exploding gradients.

**Other Comments Or Suggestions:**

- The "Impact Statement" is missing but mandatory. Could the authors please add it?
- There are parts of the code (e.g., visualization_losses.ipynb) that are not described in the paper, which is confusing. Could the authors clean the codebase such that only the relevant parts are kept?

**Other Strengths And Weaknesses:**

**Strengths**

- The paper is clear and well written
- Notations and technical background are well introduced
- I appreciate that the authors explain the context and potential impacts of each theoretical finding
- The analysis is well conducted with elegant and well-explained proofs

**Weaknesses**

I list below what I think are weaknesses, but I would be happy to be corrected if I misunderstood some important aspects of the authors' contributions.
- The motivation of the paper is not very clear to me. The authors focus on rank collapse at initialization in width but do not provide empirical validation that this hinders performance, signal propagation, or training stability in practical scenarios. Could the author elaborate on that?
- Most of the theoretical findings assume that the inputs are orthogonal, and in particular, this enables the authors to show that the attention matrix is iid Markov. Since, to the best of my knowledge, inputs are not orthogonal in practice (this is data-dependent), from my understanding, the attention matrices will not be iid Markov anymore, even at initialization, which hinders most of the author's findings. Could the authors elaborate on that? An idea to improve the submission would be to add a discussion akin to Section A.5 of [2], where the authors theoretically and experimentally motivate the assumption 3.1 of a uniform attention matrix.
- I believe the current setting is too oversimplified, which impacts the benefits of the derived insights for more practical settings. An idea to improve the submission would be to incorporate the other transformer's components in the analysis (like in [1], [2]) or at least discuss/ experiment on how they impact (or not) the current analysis.
- The authors use their theoretical findings to derive a practical fix, but they do not test it in practical settings since the experiments, even for BERT, consider isotropic inputs and consist in observing the rank but not the impact on signal propagation or training stability.
- The authors mention that their theoretical analysis resembles practical works like [3, 4]. Since the authors also provide a practical fix, more discussion on the novelty should be done, and I believe they should compare their fix with the ones in [3, 4].

Overall, the idea is interesting, and the theory is elegant with the use of random matrix tools. However, I do not find the problem well motivated enough and the analysis convincing to solve rank collapse in width given the strong assumptions needed on the input.  This is the reason for my current score, but I remain open to modifying my score, provided the authors clarify the points mentioned in the weaknesses section.

*References*

[1] Dong et al. Attention is not all you need: Pure attention loses rank doubly exponentially with depth, ICML 2021

[2] Noci et al. Signal propagation in transformers: Theoretical perspectives and the role of rank collapse, NeurIPS 2022

[3] Ye et al. Differential transformer. arXiv 2024

[4] Ali et al. Centered self-attention layers. arXiv 2023.

**Update after rebuttal**: increased score from 2 to 3.

**Questions For Authors:**

1) The author mentioned in the introduction that rank collapse in depth is not specific to attention as it is simply due to matrix multiplication. While it is clear to me that matrix multiplication can lead to gradient vanishing or exploding via the chain rule, I have trouble seeing why it naturally implies rank collapse. Could the authors provide some reference to the observation of rank collapse in weight matrices of other architectures than transformer and where it impacts the signal propagation or training stability?

2) I did not understand the author's justification in Remark 3.3 for initializing the value matrix with unit standard deviation. BERT(base model) has an embedding dimension of $768$ and a context window of $512$, meaning that $d$ and $T$, using the authors' notations, are comparable. However, in many other scenarios, one could have $d << T$, in which case the singular values of the value matrix do not compensate the scaling of the signal in all directions but one by $1/\sqrt{T}$. Could the authors clarify this point, please?

3)  [2] showed that the attention matrix at initialization resembles a uniform matrix (assumption 3.1 motivated in Section A.5 of [2]). Denoting $U = \mathbb{1}_{T \times T}$, it means that Eq. (7) of the current submission would become $\frac{1}{T} U \approx A = \frac{1}{T}U+ A^\perp$, which does not convene much meaning. What do the authors think of assumption 3.1 with respect to the simple derivation above?

4) The authors show that even with their fix, the rank collapses in depth (Fig. 4(b)). Out of curiosity, could the authors experiment (or discuss) how solutions for rank collapse in depth proposed in prior works behave on the rank collapse in width (e.g., solutions from [1, 2])?

5) Could the authors discuss more the solutions of [3, 4] and compare their fix to them?

6) The rank collapse observed in practice (Fig. 3) seems quite small (from 1.20 to 1 in stable rank) given the size of the attention matrices (512 x 512 in BERT base model). As such, it does not convince me that rank collapse in width is important to solve. Could the authors elaborate on that? An idea to improve the submission would be to see how it impacts the signal propagation or vanishing/exploding gradients in practice (akin to Fig. 1 of [2]).

*References*

[1] Dong et al. Attention is not all you need: Pure attention loses rank doubly exponentially with depth, ICML 2021

[2] Noci et al. Signal propagation in transformers: Theoretical perspectives and the role of rank collapse, NeurIPS 2022

[3] Ye et al. Differential transformer. arXiv 2024

[4] Ali et al. Centered self-attention layers. arXiv 2023.

**Relation To Broader Scientific Literature:**

I find that related work and prior works are well introduced and compared. The submission's contributions are interesting and are part of a growing interest in the literature on the theoretical understanding of transformers from a rank perspective.

**Theoretical Claims:**

The theoretical findings are supported by detailed and clear proofs.

---

> ### Author Rebuttal · Authors · 2025-04-01
>
> Thanks for your kind support of our theoretical development and for highlighting the "SAMformer" paper "SAMformer" that we looked into and will cite. For extra figures (indicated by Roman numerals), see https://anonymous.4open.science/r/spectral_analysis_transformers-C633/figures_rebuttal.pdf.
>
> +  Addressing rank collapse is motivated in multiple ways: (i) Improved quantization, crucial for compressing LLMs. Bounding the largest singular value by solving rank collapse ensures entries stay within a defined range, aiding quantization; (ii) Expressive initial representations. While it's still unclear what defines a good initialization, one might argue for avoiding rank collapse so that token representations remain diverse (as opposed to collapsing into the same vector); (iii) Controlled gradient norms. [2] shows gradients vanish after rank collapse, highlighting the importance of maintaining rank; (iv) Better generalization accuracy. [4] empirically supports our approach. While they don't directly address width, they propose subtracting the leading first-order term $\frac{1}{T} \mathbf{1}_{T\times T}$ from the attention. Their motivation lies in addressing oversmoothing in graph networks, tied to extremal eigenvalues. Their results show this subtraction increasingly benefits performance as datasets grow more complex or deeper, significantly boosting accuracy. (v) Finally, as reviewer qzvJ pointed out, our work helps bridge a theoretical gap in the understanding of attention layers in the literature.
>
> + Regarding orthogonal inputs, we refer you to our response to reviewer uxct.
>
> + We agree that including the impact of additional network modules (as in [1, 2]) is important. We do so empirically in Fig. 4, which shows rank collapse persists even with LayerNorm and skip connections. Fig. 3 further compares our theory to real-world transformers, such as BERT, which include many modules beyond attention.
>
> + Fig. 3 uses a pre-trained tokeniser, so inputs are not isometric (see response to uxct). As our work focuses on theoretical insights into attention at initialisation, we do not analyse training dynamics, which lies beyond our scope. We instead refer to [4], which offers an extensive empirical study on removing the spectral gap. Their work complements ours: they focus on experiments, while we examine the mathematical consequences of centering. Rather than replicate their study, we discuss it in the next paragraph. For signal propagation, we analyse the stable rank of token representations across depth.
>
> + Thank you for highlighting this. We’ve expanded our discussion of how our results explain [3, 4]. In [3], the authors heuristically subtract two softmax matrices, implicitly reducing energy along the dominant eigenvector. This helps mitigate rank collapse, albeit less directly and without theoretical grounding. Nonetheless, it shares the insight from [1, 2]—that rank collapse is central. Our resolution, removing the spectral outlier, has a similar effect but is rigorously understood. In [4], the same proposal of subtracting the dominant $\frac{1}{T} \mathbf{1}_{T\times T}$ term is made to “centre” attention. They show this benefits training. Our contribution is the mathematical rigour: we analyse what remains after centring and, by understanding the spectral gap, indicate what to do for other activations (e.g. ReLU, sigmoid) as shown in Fig. 1. We believe our theory supports practical efforts like [3, 4] to address rank collapse.
>
> ---
>
> Q1. Rank decreases with matrix multiplication, which leads to collapse with depth. For a training instability link, see [2] (Theorem 3.2).
>
> Q2. Please refer to the 'Scaling' paragraph in our response to uxct.
>
> Q3. Assuming uniform attention (as in [2]) removes the role of attention by treating all tokens equally. We aim to refine this by analysing the attention once the leading $\frac{1}{T}1_{T\times T}$ term is removed. Think of uniform attention as the constant in a Taylor expansion—our $A^{\perp}$ captures the next-order term.
>
> Q4. Great question. [2] proposes scaling residuals with depth to counteract collapse. But as we study rank collapse in width (even in the first layer), their fix does not apply. As shown in Fig. 4(a), modules from [1] do not significantly prevent collapse in width.
>
> Q5. Please see above.
>
> Q6. We agree with the reviewer and refer to [1] (Fig. 2), which also shows minimal changes in depth. However, we emphasise that even small changes in width can drastically affect depth. For instance, removing the spectral outlier in width causes a small change (Fig. V) but drastically alters depth behaviour (Fig. VI).
>
> ---
> [1] Dong, Y. et al. (2021). Attention is not all you need: Pure attention loses rank doubly exponentially with depth.
>
> [2] Noci, L. et al. (2022). Signal propagation in transformers: Theoretical perspectives and the role of rank collapse.
>
> [3] Ye, T. et al. (2024). Differential transformer.
>
> [4] Ali, A. et al. (2023). Centered self-attention layers.

---

> > ### Comment · Reviewer_W8Pc · 2025-04-06
> >
> > I thank the authors for their detailed answer and the additional experiments. I appreciate the authors' efforts to address my concerns. I will consider that along with the other reviews (and their responses) for my final recommendation.
> >
> > **Update**--> After carefully reading other reviews and the authors' answers to them, I decided to increase my score, given that most of my concerns are addressed. Although the setting is simplified, the analysis is well done, and additional experiments have been conducted. My main concern on the oversimplifying assumption on the data has also been addressed by the authors, which justifies the score increase.

---

> > > ### Author Response · Authors · 2025-04-07
> > >
> > > We thank you for your positive feedback and for acknowledging our efforts to address the concerns raised and conduct the requested additional experiments. If there are any remaining points that would still prevent you from recommending acceptance, we would greatly appreciate it if you could let us know, and we will do our best to address them.
> > >
> > > **Update** --> Thank you for your support!

---

### Official Review · Reviewer_qzvJ · 2025-03-18

**Overall Recommendation:** 5

**Summary:**

The paper shows that random attention layer stacks exhibit rank collapse in width (context length and latent dimension) by analyzing the spectral gap of the corresponding random matrices. They then propose a fix that replaces the attention matrix with another related matrix without spectral gap (in the limit) and show that the resulting networks do not exhibit rank collapse or gradient explosion. Experiments are done on BERT to verify the theoretical findings.

**Claims And Evidence:**

The paper claims that spectral gap in the random attention matrix at initialization causes rank collapse in width - a phenomenon well established in practice. Their proofs are sound and well-presented. The author also shows empirical evidence of their theoretical fix working on BERT-based transformers, which are popular in practice.

**Essential References Not Discussed:**

N/A

**Experimental Designs Or Analyses:**

Experiments done in the paper is well thought out and well-accommodate the theoretical claims.

**Methods And Evaluation Criteria:**

The tools used to prove the authors' propositions are well-developed from random matrix theory. Similar literature that studies transformers' rank collapse also made use of the spectrum. However, to the best of the reviewer's knowledge, this is the first work that proposes a successful and simple fix.

**Other Comments Or Suggestions:**

N/A

**Other Strengths And Weaknesses:**

Strengths:
- The paper is rigorous, yet simple. The proposed fix to rank collapse is simple and thus scalable, making the methodology generally testable even in large LLMs.


Weaknesses:
- The paper only shows rank collapse at initialization. In practice, the networks are optimized before being use. This suggests that such collapse might be a non-issue if the optimization algorithm can somehow resolve the degeneracy.
- The paper consider stacks of attention layers, while realistic transformer have more complicated structures, with layer norms and/or masking. The paper "On the Role of Attention Masks and LayerNorm in Transformers" by Wu et al. in 2024 could be a good reference to parse some of the ideas of this paper to more complicated architecture.

**Questions For Authors:**

N/A

**Relation To Broader Scientific Literature:**

Rank collapse is observed empirically in neural networks mostly due to repetitive matrix multiplication, usually referred to as oversmoothing. This phenomenon is shown to have occurred when the attention matrix is degenerate to be a uniform attention matrix, all entries sharing a numerical value. The paper argues for why the attention matrix, at initialization, would be uniform - or rank collapse in width, thus resolving a key assumption in literature.

**Theoretical Claims:**

Proofs presented in the paper is correct as far as the reviewer has examined.

---

> ### Author Rebuttal · Authors · 2025-04-01
>
> We thank the reviewer for their positive view of our work. We are happy to see that they value our contribution. We would like to address the points they raised as weaknesses:
> + The reviewer rightly points out that the direct connection between the pathological rank collapse behaviour at initialisation and the training dynamics is not yet as well established as the analogous edge-of-chaos line of research for fully connected and convolutional networks, e.g. Figs. 5 and 6 in [1]. In the case of transformers in particular, one could argue that as long as the input tokens do not entirely collapse into a single representation during the initial forward pass—i.e. if *some* information manages to propagate through the network—the early steps of the optimisation algorithm may help recover from the collapsed rank and steer training back on course. Nonetheless, empirical evidence suggests that escaping such a pathological landscape can be more difficult in practice; see Fig. 1 of [2].
>
> + While our analysis is admittedly not comprehensive of the whole transformer block, it provides valuable insights and sheds light on a hitherto overlooked phenomenon (even at the first layer) and its origin, namely rank collapse in width and softmax activation within attention layers. We also believe there is value in our results showing how the same rank collapse is impacted by changing the softmax activation to other options being advanced, such as ReLU and sigmoid. These other choices exhibit a smaller spectral gap than the softmax activation, which might explain why they are being advocated. Moreover, our resolution to subtracting the spectral gap can also be applied to ReLU and other attention activations. We believe this insight will be useful to practitioners studying alternatives. We wish to emphasise that, since rank collapse in width has a totally separate cause from its better-studied counterpart, rank collapse in depth, the ideas proposed to fix the latter do not necessarily affect the former and further distinct research is required. Thank you for the reference.
>
> + Finally, in response to some reviewers' concerns about the assumptions used in our proof of concept, we have conducted extensive ablation studies that the reviewer may find useful in confirming their positive assessment of our work. They can be found here: https://anonymous.4open.science/r/spectral_analysis_transformers-C633/figures_rebuttal.pdf
>
> --------
> [1] Schoenholz S. et al. (2016). Deep Information Propagation.
>
> [2] Noci, L. et al. (2022). Signal propagation in transformers: Theoretical perspectives and the role of rank collapse.

---

### Official Review · Reviewer_uxct · 2025-03-20

**Overall Recommendation:** 3

**Summary:**

This paper studies signal/propagation in transformers with softmax-attention at initialization. Prior work has observed rank collapse in depth (in various model architectures), which causes all tokens to converge to a single representation. It has been attributed to repeated matrix multiplications, and is known to cause issues such as exploding/vanishing gradients. This paper uncovers a phenomenon called rank collapse in width, which is unique to softmax attention layer and occurs with increasing context length. Using Random Matrix Theory, the paper shows a gap between the largest and the rest of the singular values of the attention matrix. It also shows that rank collapse in width leads to exploding gradients and exacerbates rank collapse in depth. It then proposes to mitigate rank collapse in width by removing the outlier eigenvalue(s), and empirically shows that it also helps mitigate rank collapse in depth and the issue of exploding gradients.

**Claims And Evidence:**

Yes, for the most part.

The paper clearly mentions that the focus is softmax-attention transformer models at initialization. It presents theoretical results, supporting the main claims (spectral gap leading to rank collapse in width, and connections with rank collapse in depth and exploding gradients) as well as empirical results showing improvements with the proposed approach to mitigate rank collapse.

My main concern is the strong assumptions for the results. It is stated that the results are in the regime where context length $T$ is large. However, an additional assumption that $\frac{d}{T}\in (0, 1]$ is also required for the theoretical results and is not discussed anywhere. As $T\rightarrow\infty$, the token dimension $d$ also tends to $\infty$ which seems very strong. In addition, the paper assumes that the input tokens are orthogonal, which seems unrealistic.

**Essential References Not Discussed:**

To the best of my knowledge, the paper discusses and cited the most relevant works.

**Experimental Designs Or Analyses:**

This is mainly a theoretical paper. The experimental design is fairly standard.

**Methods And Evaluation Criteria:**

The method to mitigate rank collapse makes sense. However, I have the following concern with the evaluation.

While the paper evaluates the method with standard BERT-based architecture, it should also include results with GPT-based architecture. More importantly, it should evaluate the effect of relaxing the assumption $X_0X_0^T=I$.

**Other Comments Or Suggestions:**

If you have any other comments or suggestions (e.g., a list of typos), please write them here.

- The paper should include some intuition and proof sketches for the main result. Looking through the proofs, they are not very long, and including a few key steps in the paper would be useful for the reader. For instance, in Prop. 3.4, the convergence rate is $O(T^{1-4l})$ and without the sketch, it’s unclear how this dependence on layer index $l$ arises.
- The paper should check the use of \citep and \citet.
- There are some statements that are unclear and should be rephrased. For instance, lines 59-60 (second column).

**Other Strengths And Weaknesses:**

Strength:

- The paper uncovers the phenomenon of rank collapse in width, studies it theoretically and shows connections with rank collapse in depth and vanishing/exploding gradients, which is interesting.

Weaknesses:

- The assumptions are quite strong (see Claims and Evidence section above).

- The writing should be improved, for instance, the paper doesn’t provide intuition/proof sketches for the theoretical results (see Other Comments or Suggestions section below).

**Questions For Authors:**

The paper states that “using a different scaling that makes the same quantity explode rather than vanish” (refering to gradients with respect to the value matrix $W^V_l$ at layer $l$). Can the authors elaborate on why this is the case?

**Relation To Broader Scientific Literature:**

As discussed in Section 1.1 in the paper, prior works have observed and investigated the phenomenon of rank collapse with depth in transformers as well as other architectures (at initialization). It has been linked to the issue of vanishing/exploding gradients which can disrupt training. This paper analyzes a new phenomenon, rank collapse in width, how it relates to rank collapse in depth and the issue of exploding gradients, and proposes a method to mitigate it, which also seems to resolve these other issues. It also discusses two recent works, Ye et al. 2024 and Ali et al. 2023, which indirectly and directly mitigate rank collapse and show performance improvements.

**Theoretical Claims:**

I skimmed through the proofs for the main results (Theorem 2.2, Props. 3.4 and 3.5), although I did not check them carefully.

---

> ### Author Rebuttal · Authors · 2025-04-01
>
> Thank you for your comments. Those with an editorial nature are applied in the revised draft, so we will address their main concern regarding assumptions. For additional figures (indicated by Roman numerals), please see https://anonymous.4open.science/r/spectral_analysis_transformers-C633/figures_rebuttal.pdf.
>
> **Orthonormal input tokens.** It is true that input tokens are usually not orthogonal. Nonetheless, we wish to point out that this assumption is only made for the sake of our mathematical analysis which should be seen more as a proof of concept. The emergence of spectral gap and rank collapse in width still occurs in practice without those assumptions. Let us clarify the experimental setup of Fig. 3. Sentences are tokenised with a pre-trained tokeniser, making inputs to our randomly initialised BERT clearly non-isometric. We include $\mathbf{X}_0 \mathbf{X}_0^\top$ to illustrate its deviation from identity. When tokens are instead assigned random embeddings, the covariance matrix is closer to orthogonality. Yet, the spectral gap (Figs. I(d) and II(d)) and rank collapse (Fig. 3 of the draft) persist in both cases. We have clarified this setup in our draft to emphasise this aspect. Another illustration of why orthogonality is not needed per se comes directly from Fig. 6, where the spectral gap persists in deeper layers where representations are definitely correlated. Thanks to your review, we have also added to the appendix additional plots (Fig. IX) of the spectra of a random attention layer with (synthetic) non-isometric input tokens to mirror Fig. 1.
>
> Characterising the spectrum of a generic row-normalised matrix remains an open problem in Random Matrix Theory. Our work advances the understanding of randomly initialised attention layers by assuming orthogonal input but we should acknowledge this is a limitation of our work and will emphasise this aspect in our revised version thanks to your comment. Moreover, we would like to share with the reviewer that we consciously chose to put an assumption on the input data (for our proofs to hold) to fully capture the complexity of the attention mechanism, rather than simplifying the attention mechanism as it is commonly done in the current mathematical treatment of transformers (e.g., in [1], where rank collapse is addressed under the assumption of uniform attention---essentially, no attention).
>
> **Ablation on $\gamma$.** Note that without $\gamma \leq 1$, the inputs cannot be isometric so this extra assumption was solely made for this reason. To more directly speak to your question we included new plots that explore information propagation in TinyBert, for which $d>T$ hence necessarily the tokens must be non-orthonormal. The spectral gap persists (Fig. III(a)) and rank collapse in width follows (Fig. III(b)).
>
> **Large width.** Although the theorems are formally stated as $T,d \to \infty$, we have also determined precise rates of convergence, which allows one to derive bounds in the finite case. Moreover, the convergence is sufficiently fast that, for typical practical values of $T$ and $d$ (on the order of $10^2$ to $10^3$), the quantities are already well approximated by their limiting values, see Fig. 1. In fact, it is one of the key messages we want to convey that, given the increasing scale of these architectures, asymptotic analyses (such as Random Matrix Theory) can be appropriate tools for their study.
>
> **Scaling.** As noted in Remark 3.3, we scale values as $\mathcal{N}(0,1)$ instead of $\mathcal{N}(0,1/d)$ to compensate for the softmax attention that shrinks in all but one direction. This ensures layerwise tokens remain of order one after removing the spectral gap. Alternatively, scaling by $d^{1/2}$ post-softmax would yield the same effect. Since we omit LayerNorm in our analysis, rescaling is crucial to maintain token magnitude. Note that stable rank is independent of scale so we lose no generality here. In contrast, scaling does impact gradients, so vanishing gradients in Noci et al. under traditional scaling become exploding ones in our setting.
>
> **GPT architecture.** We investigated signal propagation in a GPT2 transformer, as suggested by the reviewer (see Figs. IV(a), IV(b)). The attention matrix is now triangular due to the causal mask and its eigenvalues (real) can be read off the diagonal of this matrix. Although interesting, we feel that because it is a decoder-like architecture, an additional layer of complexity comes with considering causal masks that our theory does not cover, so this setting falls outside of our scope. Yet, we share with the reviewer the results of such an experiment and hope to maybe gain insight from them for potential follow-up work.
>
> We added a proof sketch in our revised work, thanks for the great suggestion. We hope you will consider updating your score based on our rebuttal.
>
> -----
> [1] Noci, L. et al. (2022). Signal propagation in transformers: Theoretical perspectives and the role of rank collapse.

---

> > ### Comment · Reviewer_uxct · 2025-04-08
> >
> > I thank the authors for the detailed rebuttal addressing my concerns. I have raised the score to 3.

---

> > > ### Author Response · Authors · 2025-04-08
> > >
> > > Thank you!

---

### Decision · Program_Chairs · 2025-05-01

**Decision:**

Accept (poster)

**Comment:**

This paper studies signal propagation in Transformers with Softmax attention, and demonstrates that randomly stacked attention layers exhibit rank collapse in width, by analyzing the eigenspectral gap of the corresponding random matrices.
Experimental results are provided to support the theoretical claims.

The authors have done an excellent job during the rebuttal phase, and all reviewers are now convinced of the significance and merit of the work.

As a result, I recommend acceptance of this paper to ICML 2025.
Please ensure that the additional numerical results, proof sketch, and clarifications provided during the rebuttal are included in the final version of the paper.